# Neural Temporal Walks: Motif-Aware Representation Learning on Continuous-Time Dynamic Graphs

**Ming Jin**
Monash University
ming.jin@monash.edu

**Yuan-Fang Li**
Monash University
yuanfang.li@monash.edu

**Shirui Pan** *
Griffith University
s.pan@griffith.edu.au

## Abstract

Continuous-time dynamic graphs naturally abstract many real-world systems, such as social and transactional networks. While the research on continuous-time dynamic graph representation learning has made significant advances recently, neither graph topological properties nor temporal dependencies have been well-considered and explicitly modeled in capturing dynamic patterns. In this paper, we introduce a new approach, *Neural Temporal Walks* (`NeurTWs`), for representation learning on continuous-time dynamic graphs. By considering not only time constraints but also structural and tree traversal properties, our method conducts spatiotemporal-biased random walks to retrieve a set of representative motifs, enabling temporal nodes to be characterized effectively. With a component based on neural ordinary differential equations, the extracted motifs allow for irregularly-sampled temporal nodes to be embedded explicitly over multiple different interaction time intervals, enabling the effective capture of the underlying spatiotemporal dynamics. To enrich supervision signals, we further design a harder contrastive pretext task for model optimization. Our method demonstrates overwhelming superiority under both transductive and inductive settings on six real-world datasets [1].

## 1 Introduction

Continuous-time dynamic graphs (CTDGs) consist of temporal events with respect to nodes (e.g., node addition/deletion) and edges (i.e., temporal interactions), which naturally arise in many real-world systems such as social networks and knowledge graphs [13, 32]. Traditional studies in dynamic graph modeling manually extract expressive patterns that are beneficial for understanding the crucial laws behind [36]. For example, two people are likely to know each other if they have a common friend (Figure 1). Such a dynamic graph motif describes how social connections are established [31]. Other expressive patterns, such as feedforward control loops, have also been investigated [20]. However, manually extracting motifs is expensive, time-consuming, and requires domain knowledge, thus hindering the learning on dynamic graphs with more complicated laws [36].

The advent of graph neural networks (GNNs) makes it possible to understand more complicated graphs by automatically learning the laws behind [40, 42]. While GNNs have demonstrated great success in modeling static graphs, the research on dynamic graphs is still nascent. Current research on dynamic graph neural networks (DGNNs) faces two fundamental challenges. **Firstly**, the entangled spatial and temporal dependencies in real-world CTDGs typically need a specific design to model, preventing the direct use of

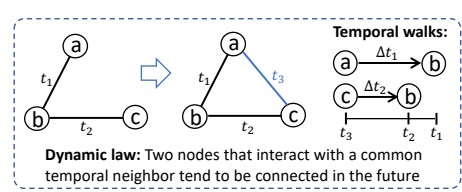

Figure 1: Temporal walks capture the law

---

*Corresponding Author.

[1]Code is available at https://github.com/KimMeen/Neural-Temporal-Walks

off-the-shelf GNNs. To overcome this barrier, previous works simplify CTDGs to a series of static graph snapshots with *uniform time intervals* (i.e., discrete-time dynamic graphs, DTDGs) [24, 29]. However, this approximation compromises the modeling precision. Although [15] and [34] propose to learn on CTDGs directly, the inductiveness of the patterns they captured is not guaranteed because node identities are directly involved in their modeling process. Some recent methods [38, 27, 36] attempt to alleviate this issue. However, they only consider time but not topological and tree traversal properties when sampling temporal neighbors, limiting their ability to extract diverse and expressive patterns from dynamic systems. **Secondly**, temporal events in CTDGs occur irregularly (e.g., nodes $a$ and $c$ interact with $b$ at different timestamps in Figure 1), resulting in a significant challenge in modeling temporal dependencies. Previous works typically bypass this challenge with the time encoding [38] to enable the use of message passing [38, 27] or sequence models [15, 36]. However, time encoding hurts model performance as temporal dependencies are modeled implicitly.

To tackle the above challenges, we propose the *Neural Temporal Walks* (`NeurTWs`) method for representation learning on CTDGs, which extracts and encodes informative dynamic graph motifs composed of irregularly-sampled temporal nodes. Specifically, motivated by [36], we propose *sptaiotemporal-biased random walks* to extract diverse and expressive patterns from a CTDG by not only considering time constrains but also topological properties and tree traversals, allowing the sampler to be better aware of the importance of temporal neighbors while maintaining the exploration and exploitation trade-off. To explicitly model temporal dependencies and capture essential dynamic laws more effectively, the extracted motifs consisting of irregularly-sampled temporal nodes are encoded with the proposed *continuous evolution* and *instantaneous activation* processes to learn time-aware node representations. The former process learns latent spatiotemporal dynamics across multiple interaction time intervals with an ordinary differential equation (ODE) function, and the second process regularizes the latent state trajectories with those irregularly-sampled observations.

On this basis, we acquire time-aware node embeddings by retrieving and encoding neighboring representative dynamic graph motifs, where a contrastive objective can be naturally designed to optimize `NeurTWs` by maximizing the mutual information between interacting temporal nodes. Compared with most existing works that use a simple link prediction objective, this harder contrastive pretext task helps enrich supervision signals, thus lifting the learning ability of `NeurTWs`.

On five benchmark datasets and a new dense e-commerce dataset, our method significantly and consistently outperforms all state-of-the-art methods in general. Specifically, it surpasses the strongest baselines by around 3% and 5% in all transductive and seven out of eight inductive link prediction tasks. It also achieves the best or on-par performance on dynamic node classification tasks. In summary, our technical contributions are three-fold: **(1)** We propose novel spatiotemporal-biased random walks to extract diverse and expressive patterns from CTDGs by considering not only time constraints but also topological and tree traversal properties; **(2)** We introduce a new perspective to encode dynamic graph motifs composed of irregularly-sampled temporal nodes, explicitly and better modeling the underlying spatiotemporal dynamics; **(3)** We integrate contrastive learning into dynamic graph modeling to enrich supervision signals, which lifts the learning ability of our model.

## 2 Related Work

**Dynamic Graph Neural Networks (DGNNs).** Existing DGNNs can be broadly classified into two categories based on their inputs. Discrete-time DGNNs operate over a sequence of evenly-sampled static graph snapshots, where different strategies are proposed to model spatial and temporal clues, e.g., combining GNNs with sequence models [30, 24, 5, 29]. Our work relates to the second category, continuous-time DGNNs, where time-dependent node or edge embeddings are learned directly on CTDGs. Among these works, an in-demand design updates latent node states by aggregating $k$-hop neighborhood information with temporal message passing. For instance, TGAT [38] samples a set of $k$-hop temporal neighbors and proposes learnable time encodings to preserve time information in message passing. TGN [27] further equips TGAT with a node memory update mechanism as in [15]. Another line of research leverages random walks to learn on CTDGs. Specifically, CTDNE [23] is the first to propose a CTDG embedding method with temporal walks. CAWs [36] extend this concept with anonymous temporal walks and uses a recurrent net to learn walk embeddings that are further aggregated when calculating interactive representations. Our method is different from prior walk-based approaches in three aspects: (1) We propose a new perspective on sampling temporal walks. While prior arts only consider time constraints, our method leverages multidimensional information,

allowing the model to explore diverse and expressive patterns from CTDGs; (2) We propose a novel and intuitive motif embedding method to model latent spatiotemporal dynamics among irregularly-sampled temporal nodes on CTDGs without relying on time encodings, which allows temporal dependencies to be modeled explicitly; (3) We replace the simple link prediction-based learning objective with a more challenging contrastive pretext task, which helps provide stronger supervision signals.

**Neural Ordinary Differential Equations (NODEs).** Chen *et al.* [2] propose a new paradigm that generalizes discrete deep neural networks by parameterizing the derivative of latent states. This concept has been applied in research areas including time series forecasting [12, 28] and computer vision [9, 1]. Recently, some works have extended NODEs to the graph domain, where most consider building deeper GNNs while alleviating the negative impacts of over-smoothing [37, 25]. Notably, the time information is absent among those works. In dynamic graph learning, most ODE-based works focus on discrete-time settings [12, 6, 8], and only a few extend NODEs to learn on CTDGs [7]. In this paper, inspired by the research on time series forecasting [28], we propose `NeurTWs` to encode extracted motifs with irregularly-sampled temporal nodes on CTDGs, which is fundamentally different from [7]: We directly integrate over multiple interaction time intervals to explicitly model the latent spatiotemporal dynamics across different temporal nodes with an ODE function, while [7] relies on a time encoding-assisted message passer to learn from historical temporal events.

**Graph Contrastive Learning (GCL).** Recently, GCL has achieved great success in graph self-supervised learning [19]. Most existing works are on static graphs [35, 43, 44, 11]. While some studies have explored the possibility of dynamic graph contrastive learning, many of them are on DTDGs. For example, STGCL [18] enhances the model's forecasting ability with DTDG augmentations and an auxiliary contrastive loss. A similar design also exists in [41, 39] and [26]. On CTDGs, Tian *et al.* [33] propose to maximize the agreement between time-aware node embeddings at different time points. In DySubC [10], the mutual information between a node and its surrounding temporal subgraphs is maximized. Different from these works, we design an effective pretext task for model optimization, where the mutual information between two nodes in an interaction is maximized. Meanwhile, we push nodes away in the embedding space if there are no temporal interactions.

## 3 Problem Formulation

We start by formally introducing the learning problem on CTDGs. A complete notation table is in Appendix A. This paper defines a CTDG as a stream of temporal interactions, i.e., $\mathcal{G} = \{(e_i, t_i)\}_{i=1}^{N}$, where each interaction has two nodes at a specific time, e.g., $(e_i, t_i) \coloneqq (\{u_i, v_i\}, t_i)$, $t_i \in \mathbb{R}^+$. As many real-world CTDG datasets are unattributed and for simplicity, we first assume these temporal interactions are without node and edge attributes and later we will discuss how our method is extended to learn on attributed CTDGs. Facing the challenge of lacking label information, DGNNs are typically supervised by temporal interactions [45]. Thus, dynamic link prediction is a widely adopted testbed to evaluate how accurate a DGNN is in predicting future interactions with the observation of historical events. Specifically, given two nodes $u$ and $v$ at time $t$ in $\mathcal{G}$, we aim to learn their time-aware embeddings $\overline{h}_u$ and $\overline{h}_v$, where the presence of an interaction between them can be predicted with a downstream classifier, i.e., $\widehat{y}_{u,v,t} = \text{clf}(\overline{h}_u, \overline{h}_v)$. The ground truth $y_{u,v,t} = 1$ if there exists an interaction between $u$ and $v$ at time $t$ otherwise $y_{u,v,t} = 0$. On this basis and with learned time-aware node representations, conducting other downstream tasks, such as dynamic node classification with another classifier, is also feasible.

## 4 The Proposed Method: Neural Temporal Walks

### 4.1 Preliminaries: Temporal Walks and Dynamic Graph Motifs

Given a dynamic graph $\mathcal{G}$, we define a motif as a subset of temporal nodes with their interactions within a defined time range [14], e.g., $0 \le t \le q$. As those motifs reflect certain dynamic laws in a CTDG, it is desirable to characterize a temporal node with its surrounding motifs.

**Definition 4.1.1** (Temporal Walk). Given a dynamic graph $\mathcal{G}$, we denote the interactions that are directly associated with a node $u$ before a cut time $t$ as $\mathcal{G}_{u,t} = \{(e, t') \mid t' < t, \, u \in e, \, (e, t') \in \mathcal{G}\}$. A (time-reversed) temporal walk rooted at node $u$ at time $t$ is defined as $W$, which is a sequence of

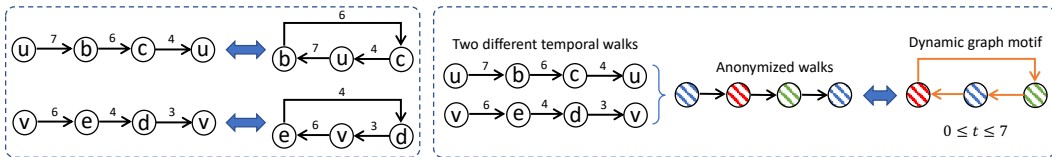

Figure 2: Triadic closures and the dynamic graph motifs: Two example temporal walks form two different triadic closures but represent the same motif within the time range $0 \leq t \leq 7$.

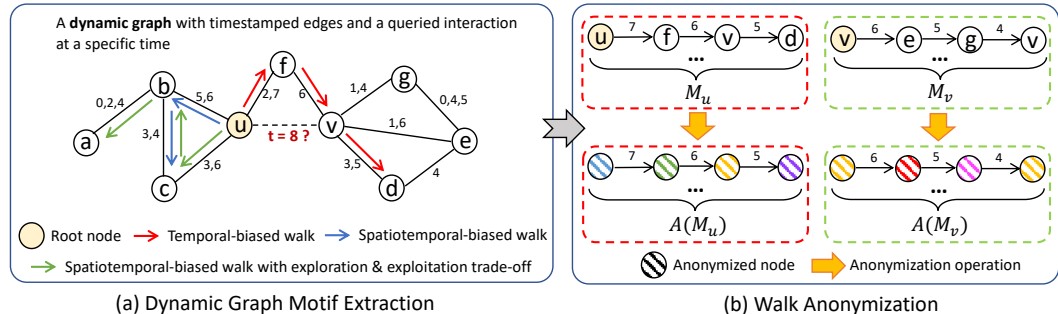

(a) Dynamic Graph Motif Extraction          (b) Walk Anonymization

Figure 3: Temporal walk sampling and anonymization. Given a CTDG and a queried interaction $(u, v, 8)$, we first extract surrounding temporal motifs by sampling a set of diverse and expressive temporal walks started from $u$ and $v$, respectively, denoted as $M_u$ and $M_v$. Then, the walks in two sets are anonymized by replacing nodes' identities with their position encodings.

temporal nodes as in [36], i.e., node $w_i$ at time $t_i$ with $w_0 := u$ and $t_0 := t$:

$$W = \{(w_i, t_i) \mid i \in \mathbb{N}, \ 0 \leq i \leq l, \ t_0 > t_1 > \cdots > t_l, \ (\{w_i, w_{i-1}\}, t_i) \in \mathcal{G}_{w_{i-1}, t_{i-1}} \text{ for } i \geq 1\}, \tag{1}$$

where $l$ is the walk length. We also use $W[i][0]$ and $W[i][1]$ (i.e., $w_i$ and $t_i$ in $(w_i, t_i)$ respectively) to denote the specific node and time in the $i$-th step.

Each walk $W$ rooted at a temporal node can actually be regarded as one of its surrounding motifs if $t_1 - t_l$ is bounded within the defined motif time range. A concrete example is given in Figure 2, where two walks in the leftmost side form two triadic closures, which essentially represent the same motif on the rightmost side after an individual walk anonymization [21]. The necessity of anonymization is to replace original node identities in walks with their relative identities, which maps each walk to a particular pattern and thus connects temporal walks and dynamic graph motifs.

**Definition 4.1.2** (Anonymous Walk). Given a temporal node $w$ and a walk $W$, the anonymization operation $A(\cdot)$ is defined as follows [36]:

$$A(w; W) = |\{v_0, \cdots, v_{i^*} \mid v_i \in W\}|, \text{ where } i^* \text{ is the smallest index s.t. } v_{i^*} = w. \tag{2}$$

## 4.2 Dynamic Graph Motif Extraction

**Temporal Walk Sampling.** Existing path-based methods mainly employ a temporal-biased sampling method when extracting dynamic graph motifs [23, 36]. Specifically, given a node $u$ at time $t$, the probability of its neighbor $a$ in $(\{a, u\}, t') \in \mathcal{G}_{u,t}$ to be sampled is proportion to $exp(\alpha(t' - t))$, which discounts stale neighbors and tends to sample more current nodes with timestamps closer to $t$. A larger $\alpha$ emphasizes more on this bias. Although more current neighbors are more likely to be informative, the underlying topological and tree traversal properties are not respected, which hinders the extraction of diverse and expressive patterns. Here, we propose *spatiotemporal-biased random walks* with the exploitation and exploration trade-off.

Our motivations are twofold: (1) Most-recent neighbors should be allocated a larger sampling probability $\propto exp(\alpha(t' - t))$ as they are typically more informative w.r.t. a root node at time $t$. In Figure 3(a), given a root node $u$ and its two temporal neighbors $f$ and $c$, a temporal-biased sampling path is more likely to be $u \rightarrow f$ instead of $u \rightarrow c$; (2) Neighbors with higher connectivity need to be emphasized to allow exploring more diverse and potentially expressive motifs. Given a node $a$ at time

$t'$, we use its degree $d_a = |\mathcal{G}_{a,t'}|$ to reify its connectivity, thus the proposed spatial-biased probability $\propto exp(-\beta/d_a)$ with a hyperparameter $\beta$ to control the bias intensity. Algorithm 1 illustrates the walk sampling procedures with the above considerations. Given a node $u$ at time $t$, the probability of its temporal neighbor $a$ to be sampled is the average of the following normalized probabilities:

$$Pr_t(a) = \frac{exp(\alpha(t_a - t))}{\sum_{a' \in \mathcal{G}_{u,t}} exp(\alpha(t_{a'} - t))} \quad (3) \qquad Pr_s(a) = \frac{exp(-\beta/d_a)}{\sum_{a' \in \mathcal{G}_{u,t}} exp(-\beta/d_{a'})} \quad (4)$$

---

**Algorithm 1** Sampling Temporal Walks

---

**Require:** Root node $w_0$, cut time $t_0$, $\mathcal{G}, C, l$
1: Initialize $\{W_c = ((w_0, t_0)) \mid 1 \leq c \leq C\}$
2: **for** $i$ in $1, 2, \cdots, l$ **do**
3:    **for** $j$ in $1, 2, \cdots, C$ **do**
4:       $w_p, t_p \coloneqq W_j[i][0], W_j[i][1]$
5:       Initialize $d_w = 0$ for all $w \in \mathcal{G}_{w_p, t_p}$
6:       **for** $(e, t)$ in $\mathcal{G}_{w_p, t_p}$ **do**
7:          Let $e \coloneqq \{w, w_p\}, d_w = |\mathcal{G}_{w,t}|$
8:       **end for**
9:       Sample one $(e, t) \in \mathcal{G}_{w_p, t_p}$ with prob. $\propto exp(\alpha(t - t_p))$ and $exp(-\beta/d_w)$
10:      Let $e \coloneqq \{w, w_p\}, W_c = W_c || (w, t)$
11:    **end for**
12: **end for**
13: **return** $\{W_c \mid 1 \leq c \leq C\}$

---

Although Algorithm 1 complements the temporal-biased schema by considering the additional topological information, it may overly encourage the depth-first search (DFS), which could misleadingly sample many homogeneous motifs with a limited budget. Take an extreme example, if $|M_u|$ is restricted to 3 in Figure 3(a), paths $u \rightarrow b \rightarrow c \rightarrow u$ may be sampled three times, leaving no room to explore $u \rightarrow c \rightarrow b \rightarrow a$ and $u \rightarrow f \rightarrow v \rightarrow d$. Thus, we design an exploitation & exploration trade-off to regularize the walk sampling with another probability:

$$Pr_e(a) = \frac{exp(-\gamma s_a)}{\sum_{a' \in \mathcal{G}_{u,t}} exp(-\gamma s_{a'})}, \quad (5)$$

where $s_a$ and $\gamma$ denotes the traversal times of node $a$ and the intensity of such a regularization.

Our complete walk sampling algorithm and its complexity analysis are in Appendices B.1 and B.3. In a nutshell, given a temporal node, the probability of its neighbor to be sampled is the average of the probabilities defined in Equations 3, 4 and 5. In NeurTWs, given a queried interaction between two temporal nodes $u$ and $v$ as shown in Figure 3, we sample a set of $C$ temporal walks rooted at each node with length $l$, denoted as $M_u$ and $M_v$.

**Anonymization.** Walk anonymization replaces node identities with position encodings (aka relative identities), which injects structural information while maintaining the inductiveness of NeurTWs. A drawback of Equation 2 is that the position encoding of each node only depends on its specific walk, leading anonymous walks rooted at the same node sharing different name spaces [36]. Thus, we consider two practical solutions: *unitary* and *binary anonymization* to address this problem. For a temporal node $w$ in at least one walk rooted at node $u$, its unitary anonmization w.r.t. $u$ considers the name space defined over $M_u$, the set of walks rooted at $u$:

$$A(w; M_u)[i] = |\{W \mid w = W[i][0], W \in M_u\}|, \text{where } i \in \{0, \cdots, l\}. \quad (6)$$

In Equation 6, the identity of $w$ is replaced by a vector $A(w; M_u)$ with length $l$, where the $i$-th element counts the number of walks that have node $w$ appearing in position $i$.

While unitary anonymization anonymizes node $w$ w.r.t. $M_u$ as $A(w; M_u)$, for interacting root nodes $u$ and $v$, $A(w; M_u)$ and $A(w; M_v)$ belong to different name spaces. Since DGNNs are typically supervised by temporal interactions, establishing the correlations between $W \in M_u \cup M_v$ may be beneficial. Thus, the binary anonymization is defined as follows [36]:

$$A(w; M_u, M_v) = A(w; M_u) || A(w; M_v), \quad (7)$$

where $||$ denotes the concatenation operation to establish the connections among motifs in $M_u$ and $M_v$. In the rest of the paper, we abbreviate the two anonymization strategies as $A(w)$ for simplicity and denote $\widehat{W} = \{(A(w_i), t_i) \mid (w_i, t_i) \in W \text{ for } i = 0, 1, \cdots, l\}$ as an anonymous walk.

## 4.3 Neural Temporal Walk Encoding

A significant challenge to model CTDGs is that interactions occur irregularly. Previous works bypass this challenge by concatenating node attributes with extra time encodings when aggregating the neighbourhood information [38, 27, 36], where temporal dependencies are modeled implicitly. A detailed discussion is in Appendix D.2. To encode a motif with irregularly-sampled temporal nodes, we explicitly integrate over multiple interaction time intervals to learn the latent spatiotemporal dynamics with those discrete observations. Figure 4 and Algorithm 2 describe our method in a nutshell, which consists of two interleaving steps: *Continuous evolution* and *instantaneous activation*.

**Algorithm 2** Neural Temporal Walk Encoding

---

**Require:** An anonymous temporal walk $\widehat{W} = \{(A(w_i), t_i) \mid (w_i, t_i) \in W \text{ for } i = 0, 1, \cdots, l\}$
1: Reverse the order of elements in $\widehat{W}$
2: $t_{-1} = t_0$, $h_{-1} = \mathbf{0}$
3: **for** i in $0, 1, 2, \cdots, l$ **do**
4:     $h_i^{'} = \text{ODESolve}(h_{i-1}, f_\theta, t_{i-1}, t_i)$
5:     $A^{'}(w_i) = \text{MLP}_\psi(A(w_i))$
6:     $h_i = g_\phi(h_i^{'}, A^{'}(w_i))$
7: **end for**
8: **return** The walk embedding $h_l$

---

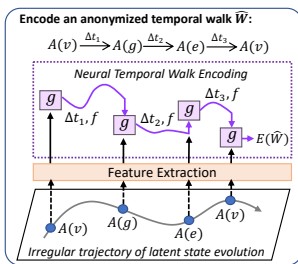

Figure 4: The spatiotemporal dynamics behind irregularly-sampled temporal nodes are explicitly modeled.

**Continuous Evolution.** Given a series of temporal nodes at different time, i.e., $(A(w_i), t_i) \in \widehat{W}$ and ensuring $t_{i-1} < t_i$ by reversing the order of elements in $\widehat{W}$, the latent spatiotemporal dynamics among those nodes are modeled as follows:

$$h_i^{'} = h_{i-1} + \int_{t_{i-1}}^{t_i} f(h_t, \theta) \, dt, \tag{8}$$

where $h_{i-1}$ denotes the latent states after encoding $(A(w_{i-1}), t_{i-1}) \in \widehat{W}$. We define the ODE function $f(h_t, \theta)$ as an autoregressive gated recurrent unit parameterized by $\theta$. See Appendix B.2.

**Instantaneous Activation.** The latent state evolution in Equation 8 conditions on a series of discrete observations. Thus, we define a function to activate latent state trajectories with instantaneous inputs:

$$h_i = g(h_i^{'}, A^{'}(w_i), \phi), \tag{9}$$

where $g(\cdot, \phi)$ can be a standard RNN cell parameterized by $\phi$, and $A^{'}(w_i) = \text{MLP}(A(w_i), \psi)$ denotes the linear mapping of a discrete observation $A(w_i)$ in an anonymous walk $\widehat{W}$.

Compared with [36], we naturally model spatiotemporal dynamics behind walks with irregularly-sampled temporal nodes, where the time information has been explicitly reflected in this modeling process. On this basis, for a temporal node $u$, we obtain its time-aware representation by pooling the embedding of walks in $M_u$, denoted as $\overline{h}_u$. In this paper, we adopt the sum pooling for simplicity.

### 4.4 Contrastive Optimization

A self-supervised pretext task is required to train DGNNs due to the scarcity of the labeling information. Most current works formulate their learning problems as a binary classification task, where the existence of an interaction is predicted. From the contrastive point of view, a binary cross-entropy loss essentially forms a particular case of the Jensen-Shannon estimator [19], where the number of negatives is one and thus provides limited supervision signals.

Here, we introduce a harder contrastive pretext task, where the mutual information between interacting temporal nodes (e.g., node $u$ and $v$ in Figure 3) is maximized. We detail the complete training algorithm of our method in Appendix B.4. In brief, the following noise-contrastive loss is minimized in the proposed approach:

$$\mathcal{L} = -\mathbb{E}\left[ \log \frac{exp\big(\text{sim}(\overline{h}_u, \overline{h}_v)\big)}{exp\big(\text{sim}(\overline{h}_u, \overline{h}_v)\big) + \sum_{v' \in \mathcal{G}, v' \neq v} exp\big(\text{sim}(\overline{h}_u, \overline{h}_{v'})\big)} \right]. \tag{10}$$

$\text{sim}(\cdot)$ measures the similarity between two entities, i.e., $\text{sim}(\overline{h}_u, \overline{h}_v) = \sigma\big(\text{MLP}(\overline{h}_u, \overline{h}_a, \xi)\big)$, where $\sigma(\cdot)$ and $\xi$ are sigmoid activation and trainable parameters.

### 4.5 Extension and Discussion

**Learning on Attributed CTDGs.** Our method can be easily extended to model CTDGs with node and interaction attributes. To achieve this, we only need to slightly modify Equation 9 as follows:

$$h_i = g(h_i^{'}, \ A^{'}(w_i) \, || \, X_{w_i} \, || \, X_{e_i}, \ \phi), \tag{11}$$

Table 1: The dataset statistics. Average interaction intensity $\lambda = 2N/(|V|T)$ [16, 36] embodies the density of interactions in a fixed period, where $T$ and $|V|$ are dataset duration and number of nodes.

| Statistics | CollegeMsg | Enron | Taobao | MOOC | Wikipedia | Reddit |
|---|---|---|---|---|---|---|
| # Nodes & Interactions | 1,899 & 59,835 | 143 & 62,617 | 64,703 & 77,436 | 7,144 & 411,749 | 9,227 & 157,474 | 10,984 & 672,447 |
| Duration (second) | 16,621,303 | 72,932,520 | 36,000 | 2,572,086 | 2,678,373 | 2,678,390 |
| # Nodes & Interaction attributes | 0 & 0 | 0 & 0 | 0 & 4 | 0 & 4 | 172 & 172 | 172 & 172 |
| Average interaction intensity $\lambda$ | $3.79 \times 10^{-6}$ | $1.2 \times 10^{-5}$ | $6.64 \times 10^{-5}$ | $4.48 \times 10^{-5}$ | $1.27 \times 10^{-5}$ | $4.57 \times 10^{-5}$ |
| # Dynamically labeled nodes | - | - | - | - | 217 | 366 |

where $X_{w_i}$ and $X_{e_i}$ are linearly mapped features of node $w_i$ and edge $e_i := \{w_{i-1}, w_i\}$ in $\widehat{W}$.

**Batching and Computational Complexity.** One computational challenge is that each walk contains irregularly-sampled nodes at different timestamps, requiring the model to separately solve $C$ different ODEs to calculate the embedding of a root node, thus hindering training in batches. To solve all ODEs at once in a minibatch with $B$ sampled interactions, we unify the integral time among all ODEs (see Appendix B.3) to a certain range, which results in a lower time complexity of $\mathcal{O}(l)$ instead of $\mathcal{O}(Bl)$. In Appendix B.3, we also provide detailed complexity analysis and discuss how to make solving Equation 8 tractable with very large time intervals.

## 5 Experiments

### 5.1 Experimental Setting

**Baselines.** Our model is compared with six state-of-the-art methods in modeling CTDGs. They can be divided into two categories based on their intrinsic mechanisms: (1) message passing-based methods, including DyRep [34], TGAT [38], and TGN [27]; (2) sequential model-based methods, including CTDNE [23], JODIE [15], and CAWs [36]. More details can be found in Appendix C.2.

**Datasets.** We evaluate model performance on six real-world datasets. CollegeMsg [17] is a social network dataset consisting of message sending activities. Enron [17] is an email communication network. Taobao [46] is an attributed user behavior dataset containing user-item interactions. MOOC [17] is an attributed network consisting of student-course interactions. Wikipedia and Reddit [15] are two bipartite interaction networks consisting of editor-page and user-post activities with rich attributive information. The dataset statistics are summarized in Table 1, and their detailed descriptions are in Appendix C.1.

**Evaluation Protocols.** For link prediction, we follow the evaluation protocols of CAWs [36] and consider two different downstream tasks for evaluation: *transductive* and *inductive* link prediction.

In *transductive* link prediction, we sort and divide all $N$ interactions in a dataset by time into three separate sets for training, validation, and testing. Specifically, the ranges of training, validation, and testing sets are $[0, N_{trn})$, $[N_{trn}, N_{val})$, $[N_{val}, N]$, where $N_{trn}/N$ and $N_{val}/N$ are 0.7 and 0.85.

In *inductive* link prediction, we use the same splits but mask a proportion of nodes (10%) and associated interactions during training, which are predicted during inference to evaluate the model inductiveness. Specifically, we first remove all interactions connected with masked nodes in the training set and then remove all interactions *not* associated with them in the validation and testing sets. In particular, we have two specific settings: (1) *New-Old* tasks require the model to predict the interactions with one unobserved (i.e., masked) and one observed nodes; and (2) *New-New* tasks aim to predict the interactions between all unobserved nodes.

For dynamic node classification, we follow the evaluation protocol in [27]. Specifically, we first obtain a model under the setting of *transductive* link prediction. Then, we train and test a separate classifier on top of this pre-trained model with the temporal nodes observed in $[0, N_{val})$ and $[N_{val}, N]$.

**Training Details.** We implement and train all models under a unified evaluation framework with the Adam optimizer. The tuning of primary hyperparameters is discussed in Appendix C.3. In solving ODEs, we use the Runge-Kutta method with the number of function evaluations set to 8 by default. For fair comparisons and simplicity, we use sum-pooling when calculating node representations in both our method and CAWs. We also test NeurTWs†, which is equipped with the binary anonymization, while NeurTWs adopts the default unitary anonymization. All methods are tuned thoroughly with nonlinear 2-layer and 3-layer perceptrons to conduct downstream link

Table 2: Transductive and inductive link prediction performances w.r.t. AUC. We use **bold font** and underline to highlight the best and second best performances. NeurTWs† is a vairant of our method with the binary anonymization.

| Task | | Method | CollegeMsg | Enron | Taobao | MOOC |
|---|---|---|---|---|---|---|
| Transductive | | JODIE [15] | $0.5846 \pm 0.038$ | $0.8714 \pm 0.011$ | $0.8477 \pm 0.015$ | $0.6815 \pm 0.014$ |
| | | DyRep [34] | $0.5297 \pm 0.042$ | $0.8632 \pm 0.013$ | $0.8462 \pm 0.012$ | $0.6195 \pm 0.018$ |
| | | TGAT [38] | $0.7528 \pm 0.004$ | $0.6592 \pm 0.012$ | $0.5400 \pm 0.005$ | $0.6750 \pm 0.035$ |
| | | TGN [27] | $0.8990 \pm 0.003$ | $0.8944 \pm 0.015$ | $0.8484 \pm 0.029$ | $0.7703 \pm 0.032$ |
| | | CAWs [36] | $0.9002 \pm 0.002$ | $0.9520 \pm 0.002$ | $0.8719 \pm 0.001$ | $0.6948 \pm 0.053$ |
| | | NeurTWs | $0.9526 \pm 0.002$ | $0.9564 \pm 0.005$ | **$0.9100 \pm 0.014$** | **$0.7756 \pm 0.031$** |
| | | NeurTWs† | **$0.9750 \pm 0.004$** | **$0.9704 \pm 0.012$** | $0.8911 \pm 0.014$ | $0.7470 \pm 0.028$ |
| Inductive | New-Old | JODIE [15] | $0.4589 \pm 0.028$ | $0.8182 \pm 0.022$ | $0.7626 \pm 0.002$ | $0.6304 \pm 0.006$ |
| | | DyRep [34] | $0.4486 \pm 0.021$ | $0.7241 \pm 0.025$ | $0.7641 \pm 0.012$ | $0.5504 \pm 0.010$ |
| | | TGAT [38] | $0.7240 \pm 0.008$ | $0.6131 \pm 0.049$ | $0.5537 \pm 0.018$ | $0.6410 \pm 0.024$ |
| | | TGN [27] | $0.8699 \pm 0.007$ | $0.7068 \pm 0.116$ | $0.8706 \pm 0.008$ | $0.6968 \pm 0.008$ |
| | | CAWs [36] | $0.8911 \pm 0.015$ | **$0.9612 \pm 0.002$** | $0.8744 \pm 0.004$ | $0.7479 \pm 0.023$ |
| | | NeurTWs | $0.9575 \pm 0.011$ | $0.9525 \pm 0.002$ | **$0.9316 \pm 0.018$** | **$0.7822 \pm 0.004$** |
| | | NeurTWs† | **$0.9699 \pm 0.010$** | $0.9566 \pm 0.007$ | $0.9037 \pm 0.013$ | $0.7772 \pm 0.006$ |
| | New-New | JODIE [15] | $0.5135 \pm 0.048$ | $0.7537 \pm 0.025$ | $0.7791 \pm 0.004$ | $0.8243 \pm 0.007$ |
| | | DyRep [34] | $0.5813 \pm 0.066$ | $0.7184 \pm 0.061$ | $0.7716 \pm 0.017$ | $0.5288 \pm 0.021$ |
| | | TGAT [38] | $0.7283 \pm 0.029$ | $0.6340 \pm 0.032$ | $0.5479 \pm 0.025$ | $0.6365 \pm 0.014$ |
| | | TGN [27] | $0.7745 \pm 0.102$ | $0.9217 \pm 0.026$ | $0.8701 \pm 0.011$ | $0.6448 \pm 0.053$ |
| | | CAWs [36] | $0.8974 \pm 0.009$ | $0.9777 \pm 0.001$ | $0.8762 \pm 0.004$ | $0.7558 \pm 0.036$ |
| | | NeurTWs | $0.9649 \pm 0.008$ | **$0.9906 \pm 0.007$** | **$0.9242 \pm 0.005$** | **$0.8329 \pm 0.010$** |
| | | NeurTWs† | **$0.9768 \pm 0.008$** | $0.9858 \pm 0.015$ | $0.9140 \pm 0.013$ | $0.8302 \pm 0.007$ |

prediction and node classification tasks, and we adopt the commonly used Area Under the ROC Curve (AUC) and Average Precision (AP) as the evaluation metrics. More implementation details can be found in Appendix C.3.

## 5.2 Results and Discussion

The link prediction performance of our method and state-of-the-art baselines are summarized in Table 2 (w.r.t. AUC) and Appendix D.1 (w.r.t. AP). As can be seen from this table, our method achieves the best performance on different tasks and datasets in general. Notably, NeurTWs surpasses the strongest model, CAWs, significantly on both unattributed and attributed CTDGs, e.g., 7.76% and 4.78% on average on CollegeMsg and Taobao. Particularly, NeurTWs and NeurTWs† achieve a strong AUC result $\geq 0.95$ on CollegeMsg, while all baselines have their AUCs $\leq 0.9$, demonstrating the effectiveness of the proposed method.

Table 3: Dynamic node classification performance w.r.t. AUC. We use **bold font** and underline to highlight the best and second best performances. The baseline results are taken from [27].

| Method | Wikipedia | Reddit |
|---|---|---|
| CTDNE [23] | $0.7589 \pm 0.005$ | $0.5943 \pm 0.006$ |
| JODIE [15] | $0.8484 \pm 0.012$ | $0.6183 \pm 0.027$ |
| DyRep [34] | $0.8459 \pm 0.022$ | $0.6291 \pm 0.024$ |
| TGAT [38] | $0.8369 \pm 0.007$ | $0.6556 \pm 0.007$ |
| TGN [27] | $0.8781 \pm 0.003$ | **$0.6706 \pm 0.009$** |
| NeurTWs | **$0.8851 \pm 0.003$** | $0.6652 \pm 0.022$ |

Specifically, on both transductive and inductive tasks, we make the following observations. (1) Our method performs well on both attributed and unattributed datasets, while the effectiveness of baseline models varies significantly. For instance, JODIE and DyRep have enormous performance gaps between Taobao and CollegeMsg in all three tasks. It indicates that our NeurTWs method is better at capturing essential dynamic laws. This superiority can be attributed to our spatiotemporal walks and associated encoding mechanism; (2) Our method is effective under both transductive and inductive settings. On the contrary, some baseline models cannot well generalize to predict interactions with unseen nodes (e.g., the performance of JODIE and DyRep drops on inductive tasks) because they rely on node identities. TGAT, TGN, and CAWs can yield better performances but leave room to improve. It is worth noting that simple unitary anonymization (i.e., NeurTWs) and more complex binary anonymization (i.e., NeurTWs†) yield similar performance in most cases. Our method with the default, simpler unitary anonymization performs significantly better than CAWs, which utilizes binary anonymization, again demonstrating the superiority of our approach in modeling

Table 4: Ablation study with the proposed `NeurTWs` method. The performance in predicting *all inductive* interactions is reported.

| No. | Configuration | CollegeMsg | | Taobao | |
|---|---|---|---|---|---|
| | | AUC | AP | AUC | AP |
| 0 | Full model (`NeurTWs`) | **0.958 ± 0.01** | **0.966 ± 0.01** | **0.938 ± 0.02** | **0.933 ± 0.02** |
| 1 | w/o T-biased probability | 0.918 ± 0.02 | 0.928 ± 0.02 | 0.932 ± 0.03 | 0.927 ± 0.01 |
| 2 | w/o S-biased probability | 0.949 ± 0.02 | 0.958 ± 0.02 | 0.915 ± 0.01 | 0.915 ± 0.01 |
| 3 | w/o E&E-biased probability | 0.957 ± 0.01 | 0.965 ± 0.01 | 0.926 ± 0.01 | 0.927 ± 0.01 |
| 4 | w/o continuous evolution | 0.868 ± 0.02 | 0.898 ± 0.01 | 0.860 ± 0.05 | 0.901 ± 0.02 |
| 5 | w/o contrasitve learning | 0.954 ± 0.01 | 0.962 ± 0.01 | 0.935 ± 0.01 | 0.932 ± 0.01 |

CTDGs without relying on sophisticated position encodings; (3) Comparing the results on Taobao, our method's improvements over CAWs are more significant than on CollegeMsg, e.g., 7.48% vs. 3.81% on transductive and 7.91% vs. 5.26% on inductive settings. Similar observations can also be found when compared on MOOC and CollegeMsg. This is because the CollegeMsg dataset has a lower average interaction intensity (i.e., $\lambda = 3.79 \times 10^{-6}$) compared with Taobao and MOOC datasets (i.e., $\lambda = 6.64 \times 10^{-5}$ and $4.48 \times 10^{-5}$). A lower $\lambda$ indicates a larger average time span in a temporal walk, for which our proposed continuous evolution process is more effective.

In addition, we compare the performance of our method (i.e., `NeurTWs`) on dynamic node classification with state-of-the-art baselines as shown in Table 3, where our approach achieves the best or on-par performances, further confirming the effectiveness of the proposal. We do not report the results of `NeurTWs`† because node classification mainly considers properties of temporal nodes rather than interactions.

## 5.3 Ablation Study

In Table 4, we report the results of our ablation study on predicting *inductive* interactions in both new-old and new-new settings. Specifically, ablations 1, 2, and 3 remove Equations 3, 4, and 5 when sampling temporal walks. Ablation 4 disables Equation 8 when encoding an anonymous walk. Ablation 5 replaces our contrastive loss with a binary cross entropy loss. From the above variants, we see performances degradation when retrieving motifs without considering the temporal or spatial information, demonstrating the effectiveness of the proposed spatiotemporal-biased walk sampler. Moreover, disabling the exploitation & exploration trade-off also hurts performances. When removing the continuous evolution, more severe performance drops are found on CollegeMsg, further demonstrating that `NeurTWs` excel on CTDGs with more sparse interactions where existing works usually fail to model them well. The performance drops also exist when our contrastive pretext task is disabled. A specific study related to continuous evolution is in Appendix D.2.

## 5.4 Parametric Sensitivity

We study the important settings in `NeurTWs` and have the following observations: (1) For each dataset, there are sweet spots in balancing three intensities (i.e., $\alpha$, $\beta$, and $\gamma$). Specifically, different datasets with different average interaction intensities, namely $\lambda$, prefer different temporal-biased intensities. For example, a relatively large $\alpha$ benefits on CollegeMsg but hurts the performance on Taobao as shown in Figure 5(a). On both datasets, overly emphasizing the topology information with a large $\beta$ hurts the performance (see Figure 5(b)) because the temporal information is overly neglected. However, setting a relatively large $\gamma$ seems beneficial (see Figure 5(c)), indicating the importance of balancing exploitation and exploration in temporal walk sampling; (2) We also investigate the choice of walk length $l$ and number of walks $C$ as shown in Figures 5(e) and 5(f). On both datasets, sampling 16 or 32 walks with the length 2 or 3 seems sufficient to characterize a temporal node; (3) For efficiency, we limit the number of negatives in Equation 10 instead of calculating $\bar{h}_{v'}$ for all $v' \in \mathcal{G}$ where $v' \neq v$. Specifically, we find that a large number is usually beneficial on both datasets as shown in Figure 5(d), where the performance increases to converge with an increasing number of negatives. In our method, we normally do not have to tune continuous evolution-related hyperparameters, but we provide a detailed study in Appendix D.2.

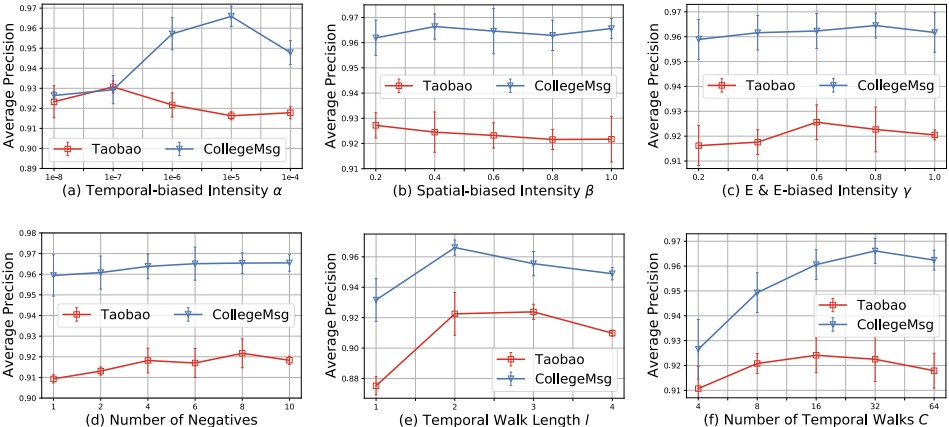

Figure 5: Study on important settings of `NeurTWs`. The performance in predicting *all inductive* interactions is reported.

## 6    Discussion and Conclusion

**Limitations.**  There are a few limitations for our method. Firstly, a more sophisticated time interval normalization is required. In real-world dynamic graphs, there may be some large time intervals between temporal nodes when constructing temporal walks. Although we propose a simple solution based on logarithmic transformations (see Appendix B.3), there is no theoretical guarantee of stability when solving the continuous evolution process by this normalization trick. Secondly, the calculation of spatial-biased probabilities is computationally heavy. While it is practical to limit the number of spanned temporal neighbors to alleviate the computational burden (see Appendix D.4), we resort to finding a more efficient implementation in the future.

**Social Impacts.**  Continuous-time dynamic graph representation learning benefits a wide range of real-world applications, including but not limited to recommender systems, social network mining, and industrial process modeling. However, there are also some potential negative impacts. For example, `NeurTWs` may learn skewed representations if there are biased patterns in the training data, which may result stereotyped predictions. Also, powerful dynamic graph models may augment harmful activities (e.g., attacking and phishing) on real-world dynamic systems.

**Conclusion.**  We propose a novel and competitive method for modeling CTDGs, which has advantages in three aspects. Firstly, our approach complements existing temporal neighborhood sampling algorithms by simultaneously considering temporal, topological, and tree traversal properties, enabling diverse and expressive dynamic graph motifs to be retrieved. Secondly, the proposed method allows these motifs with irregularly-sampled temporal nodes to be better embedded, where temporal dependencies are now explicitly modeled. Thirdly, our model forms a harder contrastive pretext task to enrich supervision signals. Consequently, the `NeurTWs` method outperforms existing works by large margins, revealing significant application prospects in many real-world scenarios, which will be an exciting focus in our future work.

## Acknowledgments and Disclosure of Funding

Some computing resources for this project are supported by MASSIVE [2]. We thank Guangsi Shi for additional computational resources and assistance in model training and deployment.

S. Pan was partially supported by an Australian Research Council (ARC) Future Fellowship (FT210100097). Y.-F. Li was partially supported by the DARPA CCU program (HR001121S0024).

---

[2] https://www.massive.org.au/

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
