# A Notations

Table 5: Summary of important notations

| Notation | Explaination |
|---|---|
| $\mathcal{G}$ | A CTDG composed of a series of temporal interactions, i.e., $\mathcal{G} = \{(e_i, t_i)\}_{i=1}^{N}$ |
| $(e, t)$ | A temporal interaction at time $t$ between nodes $u$ and $v$ where $e := \{u, v\}$ |
| $\mathcal{G}_{u,t}$ | A dynamic subgraph composed of the 1-hop temporal neighbors of node $u$ at time $t$ |
| $X_u, X_e$ | The attributes of node $u$ and interaction $e$ at a particular time |
| $W$ | A (time-reversed) temporal walk composed of a sequence of temporal nodes |
| $\widehat{W}$ | An anonymous temporal walk where node identities in $W$ are replaced by positional encodings (i.e., relative node identities) |
| $M_u$ | A set of temporal walks rooted at node $u$ at a particular time |
| $A(M_u)$ | A set of anonymous temporal walks rooted at node $u$ at a particular time |
| $h_i'$ | The latent states of the continuous evolution process at time $t_i$ when encoding $\widehat{W}$ |
| $h_i$ | The activated $h_i'$ with the instantaneous observation $A(w_i) \in \widehat{W}$ at time $t_i$ |
| $\overline{h}_u$ | The learned time-aware representation of node $u$ at a particular time |
| $A(\cdot)$ | The unitary (or binary) temporal walk anonymization |
| $f(\cdot, \theta)$ | The ODE function parameterized by $\theta$ in Equation 8 |
| $g(\cdot, \phi)$ | The activation function parameterized by $\phi$ in Equation 9 |
| $N$ | The total number of temporal interactions in $\mathcal{G}$ |
| $\alpha, \beta, \gamma$ | The intensities of temporal, spatial, and exploration & exploitation biases |
| $l, C$ | The walk length and the number of walks rooted at a temporal node |

# B Additional Model Details

## B.1 Tree-Structured Walk Sampling

In practice, we sample walks with a tree-structured sampling algorithm, where temporal, topological, and tree traversal properties are considered simultaneously (Section 4.2). In Algorithm 3, we configure the sampling with a specific hyperparameter $ngh$ that defines the sampling hops and the number of temporal neighbors to be sampled per hop. For example, we sample 32 1-hop and 1 2-hop neighbors if $ngh = [32, 1]$, which forms $C = 32$ walks with the length $l = 2$. In other word, we have $|ngh| = l$ and $\prod_{i=0}^{|ngh|-1} ngh[i] = C$.

---

**Algorithm 3** Tree-Structured Temporal Walk Sampling

**Require:** Root node $w_0$, cut time $t_0$, dynamic graph $\mathcal{G}$, sampling configuration list $ngh$, temporal-biased intensity $\alpha$, spatial-biased intensity $\beta$, exploration & exploitation-biased intensity $\gamma$
1: Initialize $roots = [(w_0, t_0)]$, $L_0 = [(w_0, t_0)]$, empty lists $L_i$ for $i$ in $1, 2, \cdots, |ngh|$
2: **for** $i$ in $1, 2, \cdots, |ngh|$ **do**
3:     $k = ngh[i]$
4:     **for** $root$ in $roots$ **do**
5:         $(w_p, t_p) = root$, initialize dictionaries $s = \{w : 0 \mid w \in \mathcal{G}_{w_p, t_p}\}$ and $d = \{w : 0 \mid w \in \mathcal{G}_{w_p, t_p}\}$
6:         $Pr_t = \{w : exp(\alpha(t - t_p)) \mid (\{w, w_p\}, t) \in \mathcal{G}_{w_p, t_p}\}$          /* *Temporal-biased probabilities* */
7:         **for** $(\{w, w_p\}, t)$ in $\mathcal{G}_{w_p, t_p}$ **do**
8:             $d[w] = |\mathcal{G}_{w,t}|$
9:         **end for**
10:        $Pr_s = \{w : exp(-\beta/d[w]) \mid w \in \mathcal{G}_{w_p, t_p}\}$          /* *Spatial-biased probabilities* */
11:        **for** $j$ in $1, 2, \cdots, k$ **do**
12:            $Pr_e = \{w : exp(-\gamma s[w]) \mid w \in \mathcal{G}_{w_p, t_p}\}$          /* *E&E-biased probabilities* */
13:            Sample one $(\{w, w_p\}, t) \in \mathcal{G}_{w_p, t_p}$ with prob. $\propto Pr_t, Pr_s, Pr_e$
14:            $L_i = L_i || (w, t)$
15:            $s[w] += 1$
16:        **end for**
17:     **end for**
18:     $roots = L_i$
19: **end for**
20: **return** $\{W_c \mid 1 \le c \le C\} = \text{Tree2Walk}(L_0, \cdots, L_{|ngh|})$          /* *Output C walks with length l* */

---

## B.2 Autoregressive Gated Recurrent Unit

Different from the gated recurrent unit (GRU) [4] whose hidden states are simultaneously activated by inputs and previous states, we define the autoregressive GRU as follows:

$$
\begin{aligned}
z_t &= \sigma(W_z h_{t-1} + b_z), \\
r_t &= \sigma(W_r h_{t-1} + b_r), \\
\widehat{h}_t &= \tanh(W_h(r_t \odot h_{t-1}) + b_h), \\
h_t &= z_t \odot \widehat{h}_t + (1 - z_t) \odot h_{t-1}.
\end{aligned}
\tag{12}
$$

In Equation 12, the hidden states $h_t$ only depend on $h_{t-1}$, where $z_t$, $r_t$, and $\widehat{h}_t$ are update, reset, and candidate activation states. Specifically, $W_z$, $W_r$, and $W_h$ are trainable weights. $b_z$, $b_r$, and $b_h$ are trainable biases. $\sigma(\cdot)$ and $\tanh(\cdot)$ denote the sigmoid and hyperbolic tangent activation functions.

## B.3 Batching and Complexity Analysis

**Walk encoding in batches** For $C$ different length-$l$ anonymous walks with various time intervals, we normally have to solve $C \times l$ different ODEs to encode each walk, which is computational expensive. Now take $C$ walks with $l = 1$ (i.e., the walks composed of only two temporal nodes) for example, we suppose the timestamps of two (i.e., source and target) nodes are $t_{start}^c$ and $t_{end}^c$ for $c$ in $1, 2, \cdots, C$. Inspired by Chen *et al.* [3], we aim to construct a *substitute variable s* for each continuous evolution process that unify their integral intervals to a specific range, e.g., 0 to 1, so that makes it possible to encode all $C$ walks at once in parallel.

Considering the $c$-th walk in the above example, we intend to integrate from $t_{start}^c$ to $t_{end}^c$ with the initial states $h_0$ and an ODE function $f(h_t, t, \theta)$ to learn the latent spatiotemporal dynamics. To integrate from 0 to 1, we transform $h_t$ to $\widetilde{h}_s$ with $s = (t - t_{start}^c)/(t_{end}^c - t_{start}^c)$. In other word, we have $t = s(t_{end}^c - t_{start}^c) + t_{start}^c$ and the following equation:

$$
\widetilde{h}_s = h_{s(t_{end}^c - t_{start}^c) + t_{start}^c}
\tag{13}
$$

The corresponding ODE function $\widetilde{f}(\widetilde{h}_s, s, \theta)$ then follows:

$$
\begin{aligned}
\widetilde{f}(\widetilde{h}_s, s, \theta) &:= \frac{d\widetilde{h}_s}{ds} = \left.\frac{dh_t}{dt}\right|_{t=s(t_{end}^c - t_{start}^c) + t_{start}^c} \frac{dt}{ds} \\
&= \left.f(h_t, t, \theta)\right|_{t=s(t_{end}^c - t_{start}^c) + t_{start}^c} (t_{end}^c - t_{start}^c) \\
&= f(\widetilde{h}_s, s(t_{end}^c - t_{start}^c) + t_{start}^c, \theta)(t_{end}^c - t_{start}^c)
\end{aligned}
\tag{14}
$$

On this basis, we have the following equation w.r.t. the facts that $\widetilde{h}_0 = h_{t_{start}^c} = h_0$ and $\widetilde{h}_1 = h_{t_{end}^c}$

$$
h_{t_{end}^c} = \text{ODESolving}(h_0, \widetilde{f}_\theta, 0, 1) = \text{ODESolving}(h_0, f_\theta, t_{start}^c, t_{end}^c),
\tag{15}
$$

where ODESolving($\cdot$) denotes a black-box ODE solver, e.g., Euler or Runge-Kutta 3/8 method. With the above reparameterization trick, we can solve $C$ different continuous evolution processes in parallel, thus allowing our method to be trained more efficiently.

**Time interval normalization** There may be some large time intervals between two temporal nodes in real-world dynamic graphs, e.g., Figure 6(a), which makes solving the continuous evolution process (i.e., Equation 8) intractable. A good idea to address this challenge is to scale time intervals without violating the inductive biases that the relative differences between time intervals reflect essential temporal clues. On this basis, various transformations can be applied, such as logarithmic transformations or divided by a large constant, to scale down the range of time intervals and make solving Equation 8 tractable. In practice, we empirically find that using the base 10 logarithmic scaling, as shown in Figure 6(c), is sufficient for our method to achieve good performances on dynamic graph modeling. Compared with other alternatives, e.g., divided by a large constant as shown in Figure 6(b), the logarithmic transformation with a properly selected base can scale down the

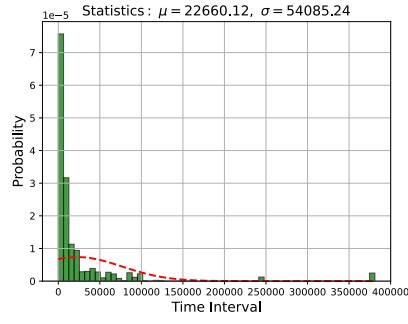

(a) Distribution of raw time intervals in seconds.

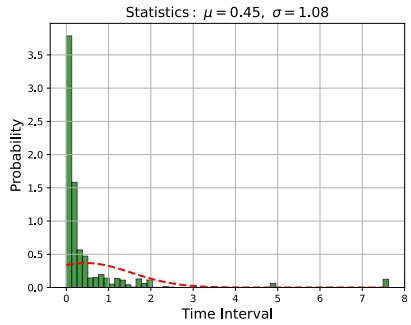

(b) Distribution after constant-based scaling with the denominator $5 \times 10^4$.

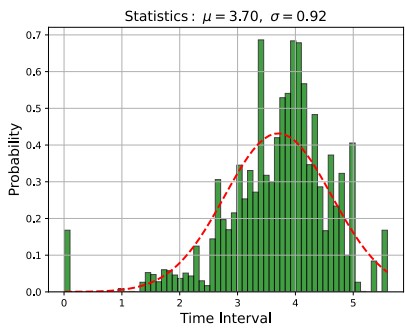

(c) Distribution after logarithmic scaling with the base 10.

Figure 6: Average time intervals between temporal nodes in a batch of walks on the Reddit dataset. Specifically, we let batch size $B = 32$ and the number of walks $C = 64$ per root node with the walk length $l = 2$.

range of time intervals while well preserving the variances between most small intervals. Although the variances between very large intervals are compressed in such a case, we conjecture the key is retaining the relative differences between small and large time intervals, which reveals critical temporal patterns in dynamic graph modeling.

**Complexity analysis**  We analyse the time complexity of the main components in `NeurTWs`, as shown in Table 6.

Table 6: Time complexity analysis

| Component | Time Complexity |
|---|---|
| Temporal-biased walk sampling | $\mathcal{O}(BC)$ |
| Spatial-biased walk sampling | $\mathcal{O}(BCk_s)$ |
| E&E-biased walk sampling | $\mathcal{O}(BC)$ |
| *Subtotal* (Algorithm 3 and anonymization) | $\mathcal{O}(BC(k_s + l))$ |
| Continuous evolution | $\mathcal{O}(lFd_h^2)$ |
| Instantaneous activation | $\mathcal{O}(ld_h^2)$ |
| *Subtotal* (Algorithm 2) | $\mathcal{O}(lFd_h^2)$ |
| **Total** (Algorithm 4) | $\mathcal{O}\big(K(BC(k_s + l) + lFd_h^2)\big)$ |

We start from analysing the time complexity of the proposed temporal walk sampling based on Algorithm 3 with the following assumptions: (1) We have a batch of $B$ interactions; (2) The sampling configuration $ngh = [k_1, k_2, \cdots, k_l]$ s.t. the number of walks $C = \prod_{i=1}^{|ngh|} k_i$ and the walk length $l = |ngh|$; (3) The time of neighborhood finding is constant; (4) The average number of spanned

temporal neighbors is $k_s$. On this basis, the time complexity of temporal-biased, spatial-biased, and exploration & exploitation-biased walk sampling are $\mathcal{O}(BC)$, $\mathcal{O}(BCk_s)$, and $\mathcal{O}(BC)$, respectively. Thus, the time complexity of our sampling algorithm is $\mathcal{O}(BCk_s)$. For walk anonymization, we analyze the upper bound time complexity based on a simple brute-force implementation. We first identify unique nodes among $C$ temporal walks with the length $l$ by traversing all nodes in walks. Then, we traverse again to acquire exact node positional encodings, thus resulting in the complexity of $\mathcal{O}(Cl)$ for processing an interaction in a batch.

We then analyse the time complexity of the proposed continuous evolution and instantaneous activation processes with the aforementioned and the following additional assumptions: (1) We instantiate $g(\cdot, \phi)$ and $f(\cdot, \theta)$ with a standard and autoregressive GRU (See Appendix B.2), respectively.; (2) We assume the input and hidden dimensions are $d_i$ and $d_h$, i.e., $A(w_i) \in \mathbb{R}^{d_i \times 1}$ and $A'(w_i), h'_i, h_i \in \mathbb{R}^{d_h \times 1}$; (3) We assume the number of function evaluations (i.e., how many times the ODE function is called in a single forward pass when solving an ODE) in the continuous evolution process is $F$. On this premise, the time complexity of solving the continuous evolution process over $B \times C$ walks is $\mathcal{O}(BClFd_h^2)$. However, the aforementioned batching mechanism allows those different ODEs to be solved in parallel, which reduces the time complexity to $\mathcal{O}(lFd_h^2)$. Regarding the instantaneous activation process, the time complexity is $\mathcal{O}(ld_h^2)$ when training in batches.

Since our model is trained by optimizing a contrastive objective function, the overall time complexity of our method is $\mathcal{O}\big(K(BC(k_s + l) + lFd_h^2)\big)$, where $K$ denotes the number of negatives.

## B.4 Model Training

---

**Algorithm 4** Training `NeurTWs`

---

**Require:** Training subgraph $\mathcal{G}_{trn}$, batch size $B$, sampling configuration $ngh$, temporal intensity $\alpha$, spatial intensity $\beta$, exploration & exploitation intensity $\gamma$, number of negatives $K$, learning rate $\eta$

1: Initialize $epoch = 1$
2: **repeat**
3:     Split all interactions in $\mathcal{G}_{trn}$ into $m$ batches $\widetilde{\mathcal{E}} = \{\widetilde{\mathcal{E}}_i \mid 1 \leq i \leq m, |\widetilde{\mathcal{E}}_i| = B\}$
4:     **for** $i$ in $1, 2, \cdots, m$ **do**
5:         Initialize $\overline{h}_{V'}^K = \{\}$
6:         */∗ TWS(·) stands for temporal walk sampling (see Algorithm 3) ∗/*
7:         Temporal walk sampling $M_U = \{M_{u_j} \leftarrow \text{TWS}(u_j, t_j, \mathcal{G}_{trn}, ngh, \alpha, \beta, \gamma) \mid (\{u_j, v_j\}, t_j) \in \widetilde{\mathcal{E}}_i\}$,
        $M_V = \{M_{v_j} \leftarrow \text{TWS}(v_j, t_j, \mathcal{G}_{trn}, ngh, \alpha, \beta, \gamma) \mid (\{u_j, v_j\}, t_j) \in \widetilde{\mathcal{E}}_i\}$
8:         Anonymization $A(M_U) = \{A(M_{u_j}) \mid M_{u_j} \in M_U\}$, $A(M_V) = \{A(M_{v_j}) \mid M_{v_j} \in M_V\}$
9:         */∗ TWE(·) stands for walk encoding (see Algorithm 2). Pool(·) denotes sum-pooling ∗/*
10:         Walk encoding and pooling $\overline{h}_U = \{\overline{h}_{u_j} \leftarrow \text{Pool}\big(\text{TWE}\big(A(M_{u_j})\big)\big) \mid A(M_{u_j}) \in A(M_U)\}$,
        $\overline{h}_V = \{\overline{h}_{v_j} \leftarrow \text{Pool}\big(\text{TWE}\big(A(M_{v_j})\big)\big) \mid A(M_{v_j}) \in A(M_V)\}$
11:         */∗ Negative sampling and calculate the time-aware node embedding of negatives ∗/*
12:         **for** $k$ in $1, 2, \cdots, K$ **do**
13:             Randomly sample a batch of nodes $V' = \{v'_j \in \mathcal{G}_{trn} \mid 1 \leq j \leq B, (\{u_j, v'_j\}, t_j) \notin \mathcal{G}_{trn}\}$
14:             Temporal walk sampling $M_{V'} = \{M_{v'_j} \leftarrow \text{TWS}(v'_j, t_j, \mathcal{G}_{trn}, ngh, \alpha, \beta, \gamma) \mid v'_j \in V'\}$
15:             Anonymization $A(M_{V'}) = \{A(M_{v'_j}) \mid M_{v'_j} \in M_{V'}\}$
16:             Walk encoding and pooling $\overline{h}_{V'} = \{\overline{h}_{v'_j} \leftarrow \text{Pool}\big(\text{TWE}\big(A(M_{v'_j})\big)\big) \mid A(M_{v'_j}) \in A(M_{V'})\}$
17:             $\overline{h}_{V'}^K = \overline{h}_{V'}^K \parallel \overline{h}_{V'}$
18:         **end for**
19:         */∗ Calculate the contrastive loss w.r.t. Equation 10 ∗/*
20:         Compute $\mathcal{L} = loss(\overline{h}_U, \overline{h}_V, \overline{h}_{V'}^K)$ and the stochastic gradients of model parameters w.r.t. $\mathcal{L}$
21:         Update model parameters w.r.t. their gradients and the learning rate $\eta$
22:     **end for**
23:     $epoch \mathrel{+}= 1$
24: **until** convergence

---

# C Experimental Setting

## C.1 Datasets

We list the introduction of six datasets used in this paper as follows. We also discuss how they are preprocessed for use in our method.

- **CollegeMsg** [3] collects the message interactions between the users in an online social media platform at the University of California, Irvine. Specifically, this dataset consists of 59,835 message interactions and 1,899 unique users, where these unattributed temporal interactions are in the format of $(u, v, t)$, denoting that user $u$ sent a message to user $v$ at time $t$. Since the format of the interactions in this dataset naturally matches the definition of continuous-time dynamic graphs in Section 3, so we do not need to preprocess the dataset intentionally. We provide the ready-to-use dataset files with our code, as well as a script to transform the raw dataset file to the data files used in our model.

- **Enron** [4] is an email network dataset consisting of around 0.5 million email interactions between about 150 senior employees in a corporation over several years. In this paper, we build the Enron dataset based on the preprocessed version in [36] by using the first half of interactions, which composes of 62,617 unattributed email interactions and 143 unique employees. We provide the preprocessed dataset files with our code.

- **Taobao** [5] is an user behavior dataset that originally collected for the research of recommendation systems, which consists of 100,150,807 user-item interactions, 987,994 users, and 4,162,024 items. Notably, each interaction in this dataset has a specific action type, i.e., a user browses/buys/favorites an item or adds it to the shopping cart. To incorporate this information, we one-hot encode each interaction's action type as the interaction attributes. In this paper, we build the Taobao dataset with a subset of the original user-item interaction network, which composes of 77,436 interactions and 64,703 unique nodes. We provide the preprocessed dataset files with our code, as well as a script to preprocess and transform the raw dataset files to the data files used in our model.

- **MOOC** [6] is an user action dataset that has a collection of actions taken by the users on the MOOC platform. This dataset consists of 7,047 users, 97 items, and 411,749 attributed interactions over the duration of about a month. Specifically, the user-item interactions in this dataset denote the access behavior of students to online course units. Similar to CollegeMsg, the format of this dataset naturally matches our definition of continuous-time dynamic graphs, so we have not preprocessed it intentionally. We provide the ready-to-use dataset files with our code.

- **Wikipedia** [7] is an user action dataset that has a set of interactions between users and wikipedia pages. This dataset records 157,474 attributed interactions in one month between 8,227 users and 1,000 pages. Specifically, an interaction denotes that an user edits a page, where the editing texts have converted into LIWC-feature vectors. Similar to CollegeMsg and MOOC, we have not preprocessed this dataset intentionally. Also, we provide the ready-to-use dataset files with the code.

- **Reddit** [8] is an user action dataset that collects one month of posts made by different users on the Reddit platform. Similar to Wikipedia, an interaction in this dataset denotes that an user makes a post on subreddits, where the texts are embedded into LIWC-feature vectors. Specifically, this dataset consists of 672,447 interactions, 10,000 most active users and 984 subreddits. We have not preprocessed this dataset intentionally, and we also provide the dataset files with the code.

---

[3] https://snap.stanford.edu/data/CollegeMsg.html
[4] http://www.cs.cmu.edu/~enron/
[5] https://tianchi.aliyun.com/dataset/dataDetail?dataId=649&userId=1
[6] http://snap.stanford.edu/jodie/mooc.csv
[7] http://snap.stanford.edu/jodie/reddit.csv
[8] http://snap.stanford.edu/jodie/wikipedia.csv

## C.2 Baselines

We list the introduction of baseline methods as follows.

- **CTDNE**[23] extends the static network embedding to dynamic graphs, where temporal random walks have been proposed with the skip-gram model to learn node representations.
- **JODIE**[15] updates latent states of two nodes in an interaction with two mutually-influenced recurrent neural networks, where the future node embedding trajectories can be estimated.
- **DyRep**[34] integrates sequence models with an attentive message passer, where 2-hop temporal neighborhood information is leveraged to learn time-aware node embeddings.
- **TGAT**[38] samples and attentively aggregates the information from $k$-hop temporal neighbors to learn time-aware node embeddings. In this method, the time information is preserved by the proposed random Fourier time encodings when conducting the message passing.
- **TGN**[27] proposes a general message passing-based framework to learn on CTDGs with a node memory updating mechanism, which inherits and combines the key designs in JODIE and TGAT.
- **CAWs**[36] learn interaction representations by encoding and aggregating anonymous temporal walks with a recurrent neural network and a pooling module.

Specifically, we detail how baseline models are tuned in link prediction tasks as follows.

- We adapt TGAT [9] to our unified evaluation framework, and mainly tune the following important hyperparameters: (1) For the degree of neighborhood sampling, we search the optimal values in $\{8, 16, 32, 64\}$; (2) We tune the model layers in $\{1, 2, 3\}$; (3) We use the default product attention and tune the number of attention heads in $\{2, 4, 6\}$; (4) For the dimensions of node and time embeddings, we search the optimal configuration in $\{16, 32, 64, 100\}$. Regarding the rest of hyperparameters, we use the default setting in the code of TGAT.
- For JODIE, DyRep, and TGN, we adapt the implementation [10] to our unified evaluation framework. Specifically, we mainly follow the default setting when executing JODIE on our datasets, where the dimension of node embeddings is searched in $\{32, 64, 128, 256\}$. For DyRep, we tune the degree of neighborhood sampling in $\{10, 16, 32, 64\}$ and the number of graph attention layers in $\{1, 2, 3\}$. We also search the optimal values of these two hyperparameters in TGN, where additional hyperparameters are tuned as well: (1) We tune the number of attention heads in $\{2, 4, 8\}$; (2) We tune the dimension of node, time, and message embeddings in $\{16, 32, 64, 100\}$; (3) We tune the dimension of node memories in $\{4, 16, 32, 64, 172\}$.
- Our method shares the same evaluation pipeline with CAWs [11], for which we mainly tune the following hyperparameters based on their default setting: (1) We tune the walk length in $\{1, 2, 3\}$ and the number of walks in $\{16, 32, 64, 128\}$; (2) We tune the time decay in $\{10^{-7}, 10^{-6}, 10^{-5}, 10^{-4}\}$; (3) We tune the model with the unitary and binary anonymization, where the best performances are reported. We follow their default setting for the rest of the hyperparameters except for the walk pooling, which is set to summation as same as in our method for simplicity and fair comparison.

## C.3 Implementation Details

Our code is available at `https://github.com/KimMeen/Neural-Temporal-Walks`, where we provide detailed instructions for dataset preparation and model training.

We first introduce the general setting of `NeurTWs` when training on all three datasets. Specifically, our models are trained with the learning rate $10^{-4}$ and batch size 32. We set the maximum number of epochs to 50, but the model training usually converges and is early stopped within the first 20 epochs.

For important hyperparameters controlling the walk sampling and contrasitve learning, we summarize their grid search ranges in Tables 7 and 8.

---

[9] `https://github.com/StatsDLMathsRecomSys/Inductive-representation-learning-on-temporal-graphs`
[10] `https://github.com/twitter-research/tgn`
[11] `https://github.com/snap-stanford/CAW`

Table 7: Search ranges of important hyperparameters on two unattributed datasets

| Hyperparameter | CollegeMsg | Enron |
|---|---|---|
| Walk length $l$ | $\{1, 2, 3\}$ | $\{1, 2, 3, 4\}$ |
| Number of walks $C$ | $\{16, 32, 64\}$ | $\{16, 32, 64\}$ |
| Number of negatives | $\{1, 3, 6, 9, 18\}$ | $\{2, 4, 6, 8, 16\}$ |
| Temporal bias intensity $\alpha$ | $\{10^{-6}, 10^{-5}, 10^{-4}\}$ | $\{10^{-7}, 10^{-6}, 10^{-5}\}$ |
| Spatial bias intensity $\beta$ | $\{0, 10^{-3}, 10^{-2}, 10^{-1}, 1\}$ | $\{0, 10^{-3}, 10^{-2}, 10^{-1}, 1\}$ |
| E&E trade-off intensity $\gamma$ | $\{0, 10^{-3}, 10^{-2}, 10^{-1}, 1\}$ | $\{0, 10^{-3}, 10^{-2}, 10^{-1}, 1\}$ |

Table 8: Search ranges of important hyperparameters on two attributed datasets

| Hyperparameter | Taobao | MOOC |
|---|---|---|
| Walk length $l$ | $\{1, 2, 3\}$ | $\{1, 2, 3\}$ |
| Number of walks $C$ | $\{16, 32, 64\}$ | $\{16, 32, 64\}$ |
| Number of negatives | $\{2, 4, 6, 8, 16\}$ | $\{2, 4, 6, 8\}$ |
| Temporal bias intensity $\alpha$ | $\{10^{-7}, 10^{-6}, 10^{-5}\}$ | $\{10^{-6}, 10^{-4}\}$ |
| Spatial bias intensity $\beta$ | $\{0, 10^{-3}, 10^{-2}, 10^{-1}, 1\}$ | $\{10^{-1}, 5 \times 10^{-1}, 1\}$ |
| E&E trade-off intensity $\gamma$ | $\{0, 10^{-3}, 10^{-2}, 10^{-1}, 1\}$ | $\{5 \times 10^{-1}, 1\}$ |

For the rest of hyperparameters, we set them as follows: (1) We use the fixed-step ODE solver (i.e., Runge-Kutta 3/8 method) with the step size $= 0.125$ by default without specific tuning. Adaptive solvers (e.g., Dormand–Prince method) can also be used in our method; (2) The hidden dimensions of the position encoding has not tuned and set to 108 by default; (3) For the walk pooling, we use the summation by default for simplicity. More details can be found in our code, where we provide the scripts with the default settings to run experiments on three datasets.

All experiments are conducted on a Linux cluster with four computing nodes (for each we have $4\times$ RTX6000 24GB, $1\times$ Intel Xeon Silver 4214R, and 175 GB system memory) and a Linux server ($4\times$ GeForce RTX 2080 Ti 11GB, $1\times$ Intel Core i9-9900X, and 94GB system memory).

# D  Additional Experimental Results

## D.1  Performances in Average Precision

See Table 9 for detailed transductive and inductive link prediction results w.r.t. AP.

## D.2  Study on Continuous Evolution Process

We study the effectiveness of the proposed continuous evolution process in modeling temporal dependencies when encoding anonymous temporal walks, and further compare it with other two widely adopted strategies, i.e., exponential decay [22] and random Fourier time encoding [38]. We summarize the results in Table 10. Specifically, we use GRU as the RNN backbone without considering the time information associated with each temporal node in $\widehat{W}$ (i.e., Standard RNN). On this basis, we equip GRU with the exponential decay mechanism (i.e., RNN with exponential decay), which is defined as follows based on Equation 9 in the main body of the paper:

$$h_i = \text{GRUCell}(h_{i-1} \cdot exp(-\tau\Delta_t), A'(w_i), \phi), \tag{16}$$

where $\Delta_t = t_i - t_{i-1}$ and $\tau$ is a hyperparameter to control the rate of decay. In our experiments, we tune $\tau$ in $\{10^{-9}, 10^{-8}, 10^{-7}\}$. Apart from this, time encoding is another strategy to incorporate the time information, which is defined below [38]:

$$h_i = \text{GRUCell}(h_{i-1}, A'(w_i) \,||\, \langle\Phi_d(t_{i-1}), \Phi_d(t_i)\rangle, \phi),$$

$$\Phi_d(t) = \sqrt{\frac{1}{d}}\big[\cos(\omega_1 t)\sin(\omega_1 t), \cdots, \cos(\omega_d t)\sin(\omega_d t)\big], \tag{17}$$

where $\Phi_d(\cdot)$ denotes the time encoding with the dimensions of $d$. $\langle\cdot\rangle$ and $||$ are inner product and concatenation operation. We implement the time encoding-based variant (i.e., RNN with time encoding) based on Equation 17.

Table 9: Transductive and inductive link prediction performances w.r.t. AP. We use **bold font** and underline to highlight the best and second best performances. NeurTWs† is a vairant of our method with the binary anonymization.

| Task | | Methods | CollegeMsg | Enron | Taobao | MOOC |
|---|---|---|---|---|---|---|
| Transductive | | JODIE [15] | $0.5329 \pm 0.029$ | $0.8467 \pm 0.023$ | $0.8412 \pm 0.013$ | $0.6386 \pm 0.021$ |
| | | DyRep [34] | $0.5029 \pm 0.023$ | $0.8237 \pm 0.022$ | $0.8397 \pm 0.022$ | $0.5723 \pm 0.005$ |
| | | TGAT [38] | $0.7212 \pm 0.005$ | $0.6893 \pm 0.008$ | $0.5598 \pm 0.009$ | $0.6704 \pm 0.032$ |
| | | TGN [27] | $0.9007 \pm 0.003$ | $0.8876 \pm 0.016$ | $0.8677 \pm 0.017$ | $0.7347 \pm 0.036$ |
| | | CAWs [36] | $0.9255 \pm 0.001$ | $0.9437 \pm 0.002$ | $0.8829 \pm 0.001$ | $0.7011 \pm 0.062$ |
| | | NeurTWs | $0.9612 \pm 0.001$ | $0.9503 \pm 0.008$ | **$0.9030 \pm 0.009$** | **$0.7593 \pm 0.027$** |
| | | NeurTWs† | **$0.9773 \pm 0.003$** | **$0.9664 \pm 0.010$** | $0.8817 \pm 0.012$ | $0.7228 \pm 0.033$ |
| Inductive | New-Old | JODIE [15] | $0.4413 \pm 0.052$ | $0.8190 \pm 0.024$ | $0.8202 \pm 0.005$ | $0.6914 \pm 0.010$ |
| | | DyRep [34] | $0.4088 \pm 0.034$ | $0.7322 \pm 0.018$ | $0.8237 \pm 0.008$ | $0.5633 \pm 0.014$ |
| | | TGAT [38] | $0.6965 \pm 0.006$ | $0.6324 \pm 0.033$ | $0.5738 \pm 0.021$ | $0.6408 \pm 0.014$ |
| | | TGN [27] | $0.8632 \pm 0.002$ | $0.7046 \pm 0.128$ | $0.8792 \pm 0.013$ | $0.7271 \pm 0.002$ |
| | | CAWs [36] | $0.9179 \pm 0.014$ | **$0.9577 \pm 0.005$** | $0.8960 \pm 0.007$ | $0.7602 \pm 0.022$ |
| | | NeurTWs | $0.9653 \pm 0.010$ | $0.9466 \pm 0.003$ | **$0.9294 \pm 0.012$** | **$0.7730 \pm 0.006$** |
| | | NeurTWs† | **$0.9721 \pm 0.008$** | $0.9488 \pm 0.009$ | $0.9063 \pm 0.011$ | $0.7617 \pm 0.008$ |
| | New-New | JODIE [15] | $0.4427 \pm 0.091$ | $0.7332 \pm 0.028$ | $0.8452 \pm 0.014$ | **$0.8263 \pm 0.015$** |
| | | DyRep [34] | $0.5502 \pm 0.102$ | $0.5966 \pm 0.091$ | $0.8423 \pm 0.019$ | $0.4484 \pm 0.047$ |
| | | TGAT [38] | $0.6983 \pm 0.036$ | $0.6593 \pm 0.030$ | $0.5660 \pm 0.024$ | $0.6324 \pm 0.012$ |
| | | TGN [27] | $0.7568 \pm 0.112$ | $0.9418 \pm 0.016$ | $0.8949 \pm 0.019$ | $0.6547 \pm 0.065$ |
| | | CAWs [36] | $0.9243 \pm 0.010$ | $0.9697 \pm 0.003$ | $0.9005 \pm 0.005$ | $0.7688 \pm 0.037$ |
| | | NeurTWs | $0.9702 \pm 0.005$ | **$0.9853 \pm 0.013$** | **$0.9252 \pm 0.002$** | $0.8251 \pm 0.017$ |
| | | NeurTWs† | **$0.9780 \pm 0.008$** | $0.9845 \pm 0.016$ | $0.9133 \pm 0.014$ | $0.8067 \pm 0.013$ |

Table 10: Study on different strategies to model temporal dependencies in temporal walk encoding (Section 4.3). The performance in predicting *all inductive* interactions is reported.

| Configuration | CollegeMsg | | Taobao | |
|---|---|---|---|---|
| | AUC | AP | AUC | AP |
| Standard RNN | $0.868 \pm 0.02$ | $0.898 \pm 0.01$ | $0.860 \pm 0.04$ | $0.901 \pm 0.02$ |
| RNN with exponential decay | $0.915 \pm 0.03$ | $0.925 \pm 0.03$ | $0.923 \pm 0.01$ | $0.920 \pm 0.01$ |
| RNN with time encoding | $0.910 \pm 0.02$ | $0.903 \pm 0.01$ | $0.889 \pm 0.01$ | $0.906 \pm 0.02$ |
| Continuous evolution | **$0.958 \pm 0.01$** | **$0.966 \pm 0.01$** | **$0.938 \pm 0.02$** | **$0.933 \pm 0.02$** |

From the results in Table 10, we have the following observations: (1) All variants and our method improve the performance of standard RNN, demonstrating the importance of leveraging time information when learning on CTDGs; (2) The proposed continuous evolution process significantly surpasses the exponential decay and time encoding approaches, proving its effectiveness in capturing the temporal dependencies; (3) Although time encoding helps the model be aware of critical time information, its effectiveness is limited compared with our method.

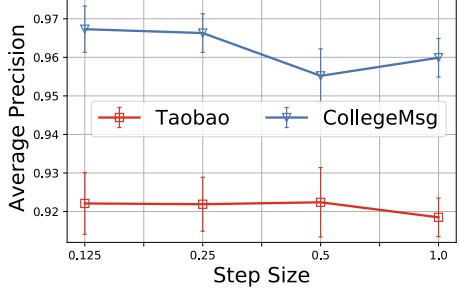

Figure 7: The performance w.r.t. different step sizes in solving the continuous evolution process. The average precision in predicting *all inductive* interactions is reported.

Although the continuous evolution process is less tuned in our method as mentioned in Appendix C.3, we study how the `NeurTWs` method is affected w.r.t. the number of function evaluations (i.e., the number of times the ODE function is called in a single forward pass when solving an ODE). The experimental results are in Figure 7, where we use the Euler method with different step sizes for illustration purposes. Notably, the Euler method is a fixed-step solver, and all ODEs to be solved are restricted to a specific integral interval (i.e., 0 to 1, see Appendix B.3). Thus, the number of function evaluations depends only on the step size and is inversely proportional to it. From the results in Figure 7, we find that the model performance decreases slightly as the step size increases when solving ODEs. This phenomenon makes sense, as a smaller step size equates to more function evaluations, which allow for more precise learning of the underlying spatiotemporal dynamics in dynamic graph motifs, but at the cost of higher complexity. In other words, our method allows a trade-off between model accuracy and efficiency.

### D.3   Visualization

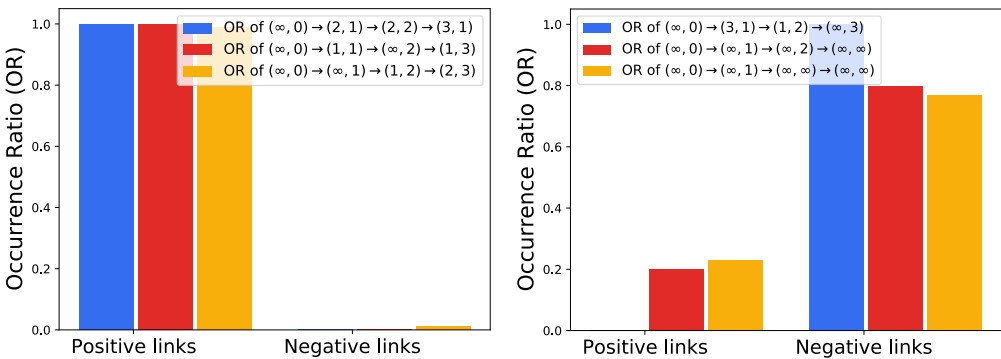

(a) The occurrence ratio of top-3 highest-scored motifs in positive and negative samples.

(b) The occurrence ratio of top-3 lowest-scored motifs in positive and negative samples.

Figure 8: Visualizing the most discriminative dynamic graph motifs learned by `NeurTWs†` on the CollegeMsg dataset, where their occurrence ratios in positive and negative samples are compared.

We follow the method described in [36] to visualize the most discriminative dynamic graph motifs learned by our method with the binary anonymization (i.e., `NeurTWs†`) on the CollegeMsg dataset.

We first illustrate how a motif can be represented in a sequence of coordinates, which essentially characterizes its shape in a coordinate system w.r.t. the extracted temporal walks rooted at two nodes of an interaction. Specifically, given a sampled interaction $(\{u, v\}, t)$ from the dynamic graph $\mathcal{G}$, we sample two sets of temporal walks $M_u$ and $M_v$ rooted at node $u$ and $v$, respectively. Recall the binary anonymization defined in Section 4.2, $A(w; M_u)[i]$ and $A(w; M_v)[i]$ count the number of walks in $M_u$ and $M_v$ that have node $w$ appearing in position $i$, where $i \in \{0, \cdots, l\}$. On this basis, we have the shortest path between node $w$ and a root node $u$ (w.r.t. the subgraph composed of the walks in $M_u$) defined as $d_{wu} := \min\{i | A(w; M_u)[i] > 0\}$. We can define $d_{wv}$ in a similar way with $A(w; M_v)$. Thus, the coordinate mapping $A(w; M_u, M_v) \to (d_{wu}, d_{wv})$ can be viewed as a way to map the position encoding of node $w$ to its coordinate in a coordinate system defined over $M_u$ and $M_v$. As a result, we can obtain the shapes of anonymous temporal walks by mapping each node's position encoding to its relative coordinate. Specifically, $\infty$ in a coordinate, e.g., $(\infty, 0)$, denotes that the target node of an interaction does not appear in the temporal walks rooted at the source node of this interaction. A special case $(\infty, \infty)$ denotes a nonexistent node, which is used in walk padding to make sure all walks have the same length $l$ when there are no temporal neighbors can be sampled.

We then describe how most discriminative motifs can be identified. Given a sampled interaction $(\{u, v\}, t)$ from the dynamic graph $\mathcal{G}$ and a well-trained model of `NeurTWs†` with a linear downstream link predictor $\Psi^\mathsf{T}(\bar{h}_u \| \bar{h}_v) := \Psi^\mathsf{T} \sum_{c=1}^{2C} \text{enc}(\widehat{W}_c)$ for $\widehat{W}_c \in A(M_u) \cup A(M_v)$, we can obtain the predictive score of each motif $\widehat{W}_c$ that contributes to the final prediction of this interaction, i.e., $\Psi^\mathsf{T}\text{enc}(\widehat{W}_c)$. In our experiments, we visualize the dynamic graph motifs with the top-3 highest and

lowest predictive scores in the testing set of the CollegeMsg dataset, as well as their normalized occurrence ratios in all positive and negative samples during the model inference.

From the results in Figure 8, we have the following observations: (1) A general dynamic law can be observed among the shapes of highest-scored motifs; that is, two temporal nodes surrounded by some common dynamic graph motifs are more likely to interact. For example, the shape of the highest-scored motif $(\infty, 0) \rightarrow (2, 1) \rightarrow (2, 2) \rightarrow (3, 1)$ indicates that two temporal nodes share some common 3-hop temporal neighbors because there is no $\infty$ in the second to fourth positions of the shape of this motif. On the contrary, two temporal nodes with few overlapped dynamic graph motifs are unlikely to interact, e.g., $(\infty, 0) \rightarrow (\infty, 1) \rightarrow (\infty, 2) \rightarrow (\infty, \infty)$. Thus, our method allows informative dynamic graph motifs and the underlying dynamic laws to be extracted and captured effectively; (2) These three highest-scored motifs commonly appear in positive samples where interactions exist between temporal node pairs but rarely appear in negative samples (see Figure 8(a)). In contrast, for those lowest-scored motifs, they appear more frequently in negative samples (see Figure 8(b)). Therefore, the discriminative power of our method is proven to be strong, which captures essential dynamic laws to characterize temporal nodes where downstream tasks can be effectively conducted.

### D.4 Complexity Evaluation

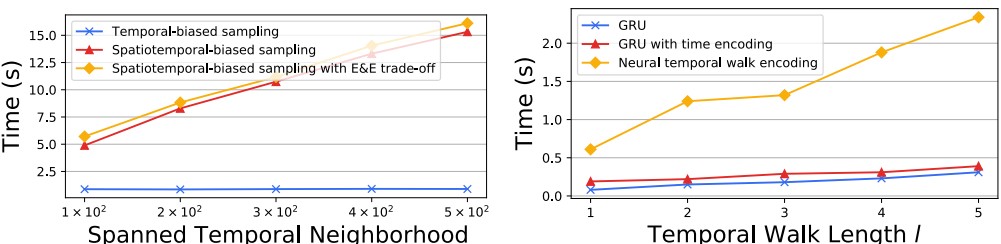

(a) Average walk sampling runtime in a batch w.r.t. the number of spanned temporal neighborhood.

(b) Average walk encoding runtime in a batch w.r.t. the temporal walk length.

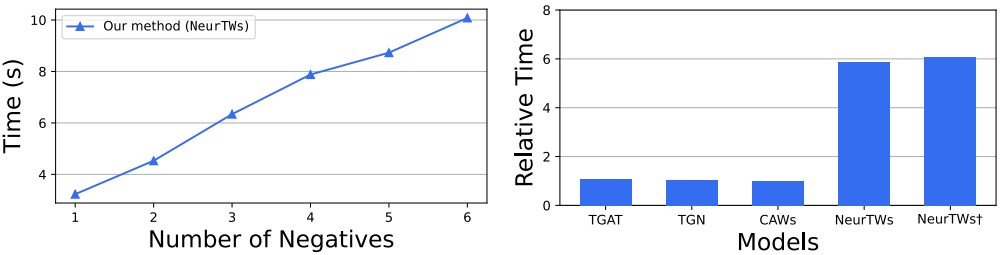

(c) Average runtime of our method in a batch w.r.t. the number of negatives.

(d) Related average runtime of different methods per batch based on their optimized settings.

Figure 9: Study the efficiency of our method from different perspectives on the Enron dataset.

We empirically study the efficiency of our method by recording and comparing the runtime of main components in temporal walk sampling and encoding. We also plot the average runtime of `NeurTWs` with different number of negatives during the optimization and the relative runtime comparison between it and other baselines under their optimized settings. The experimental results are in Figure 9, which is consistent with our complexity analysis in Appendix B.3.

We first compare the runtime of the proposed spatiotemporal-biased walk sampling with its two ablation variants on the Enron dataset under the different number of spanned temporal neighbors per node, as shown in Figure 9(a). Specifically, we use the default model setting on Enron that samples 64 walks with the length 2, but the number of negatives is set to 1 for a more straightforward comparison. The number of spanned temporal neighbors is defined as $|\mathcal{G}_{w_p, t_p}|$ in Algorithm 3. From the results in Figure 9(a), we have three observations: (1) Temporal-biased walk sampling has a constant runtime w.r.t. the number of spanned temporal neighbors. This observation can be confirmed by our complexity analysis summarized in Table 6, where the temporal-biased sampling is only in

proportion to the batch size and the number of walks to be sampled; (2) The spatiotemporal-biased walk sampling is in proportion to the number of spanned temporal neighbors, which proves our analysis in Appendix B.3 that its time complexity is $\mathcal{O}(BCk_s)$, where $k_s$ is the x-axis of Figure 9(a); (3) Our spatiotemporal-biased walk sampling with the exploration & exploitation trade-off has a similar time complexity compared to its variant without considering this trade-off. In general, although the calculation of the proposed spatial-biased probabilities induces higher time complexity when sampling temporal walks, it helps extract diverse and expressive dynamic graph motifs, e.g., disabling this mechanism hurts the performance drastically on the Taobao dataset (e.g., ablation 2 in Table 4). To alleviate the computational burden when modeling dynamic graphs with a large average number of spanned temporal neighbors, we find it is practical to limit $k_s$ by selecting a set of (potentially informative) candidates to explore, e.g., we may choose these candidate neighbors randomly or based on prior knowledge. A concrete example is Enron with a higher $k_s$ than other datasets; we empirically find that limiting the number of first- and second-order neighbors with small upper bound (e.g., 64×5 and 1×8) in a temporal-biased manner does not affect the performances of our method significantly.

We then compare the runtime of the proposed neural temporal walk encoding with one of its ablation variants (i.e., GRU, which essentially forms the instantaneous activation process without the continuous evolution process) and an in-demand design (i.e., GRU with time encoding, which adopts random Fourier time encodings as a part of input features of GRU to preserve the time information when learning walk embeddings). The experimental results are in Figure 9(b), where we use the setting as mentioned above except for the walk length, which is the x-axis of this plot. Specifically, we find that the runtime of all methods is in proportion to the walk length, which matches the complexity analysis in Appendix B.3. Compared to GRU, the proposed neural temporal walk encoding has a higher time complexity which is also in proportion to the number of function evaluations $F$ in the continuous evolution process (see Table 6). In Figure 9(b), we find that the runtime of our method is about 7.8 times that of the GRU, which is close to our training setting where $F = 8$. Although the proposed continuous evolution process results in a higher runtime of our method in this example compared with the time encoding approach, it demonstrates significantly better capabilities in digesting the time information (see Table 10). Besides, the proposed continuous evaluation process supports the trade-off between model accuracy and efficiency, allowing our encoding module to have the same time complexity as the time encoding-based GRU (i.e., $\mathcal{O}(ld_h^2)$ by letting the step size $= 1$) without compromising too much modeling precision (see Figure 7).

In Figure 9(c), we display the runtime of the entire model in a batch w.r.t. the number of negatives, which proves our justification in Appendix B.3 that the time complexity of our method is in proportion to the number of negatives during the optimization under a certain model configuration (see Table 6).

In Figure 9(d), we compare the relative runtime between our method and other baselines under their optimized settings on Enron. Although the core components in NeurTWs induce higher time complexity, they significantly improve existing approaches from various aspects in terms of the modeling precision. It is worth noting that our method explicitly supports trading precision for speed to alleviate the computational burden. For example, limiting the number of spanned temporal neighbors is practical to reduce runtime (see Figure 9(a)) without losing too much modeling precision, as discussed above. Also, reducing the number of function evaluation $F$ (e.g., increasing the step size when solving Equation 8 with a fixed-step solver) brings significant speedups, which reduces the complexity of neural temporal walk encoding (see Table 6). For the optimization, limiting the number of negatives reduces the model runtime as well.