# OpenReview forum: "Neural Temporal Walks: Motif-Aware Representation Learning on Continuous-Time Dynamic Graphs"
_NeurIPS.cc/2022/Conference — NeurIPS 2022 Accept_

### Official Review · Reviewer_avBg · 2022-07-10

**Rating:** 7
**Confidence:** 4
**Soundness:** 3 good
**Presentation:** 3 good
**Contribution:** 3 good

**Summary:**

The authors propose a method based on random walks and neural ordinary differential equations to capture and encode irregularly-sampled walks consisting of temporal nodes. The authors also provide in-depth analysis to reveal the connections between temporal graph motifs and temporal walks and why it is essential to capture them to learn temporal graph representations. Compared with previous work, this paper demonstrates good performance in both attributive and non-attributive scenarios.

**Questions:**

- How the proposed method deals with large time intervals compared with the time encoding approach?

- How to deal with the case where the number of the spanned neighborhood is large? Based on the complexity analysis in the appendices, the time complexity of the proposed sampling algorithm is proportion to $k_s$, which makes the proposed method hard to scale on dynamic graphs with higher average temporal node degree.


**Limitations:**

- The discussions of how the proposed method can learn on dynamic graphs with large average time intervals and higher temporal node degrees are necessary.

- The details of paper writing can be further improved, e.g., u->b in Fig. 3 can also be a temporal-biased walk since the timestamp of node b is the same as node f.


**Strengths And Weaknesses:**

Pros:

The paper is overall well-written, and the motivations are justified clearly. In general, the improvements over existing work on dynamic graph learning are significant based on the experimental results.

- Representation learning on continuous-time dynamic graphs is an important but less explored task. This paper presented an effective method for solving this problem.

- Overall, the novelty of the proposed method is somewhat significant compared with existing random walk-based methods. Notably, capturing the information of time intervals with NODE demonstrates significant improvements over the widely adopted time encoding approach, which may open a new possibility in modeling continuous-time dynamic graphs.

Cons:

- Regarding the walk anonymization to establish the structural connections between walks, there is no significant breakthrough compared with CAW.

- The complexity analysis is prone to error, e.g., the time complexity of walk anonymization has not been counted when calculating the total time complexity of the proposed method.

- Optimizing Eq. 8 may be problematic. In real-world dynamic graphs, the time intervals may be extremely large (in seconds), which leads to large integral intervals when solving the ODE.

---

> ### Author Response · Authors · 2022-08-02
> **Response to Reviewer avBg**
>
> We thank the reviewer for the valuable feedback, and we especially appreciate the reviewer for acknowledging the novelty and significance of our method on continuous-time dynamic graph modeling. Regarding the raised concerns, we address them as follows.
>
> - For the argument that there is no significant breakthrough in the walk anonymization compared with CAWs, we wish to clarify that this has not been claimed as one of our contributions in the paper. Although walk anonymization is essential to establish the connections between temporal walks to form dynamic graph motifs, our research aims to provide thoughtful insights to address some fundamental limitations in the previous node-based (e.g., TGAT) and subgraph-based (e.g., CAWs) dynamic graph neural networks. In a nutshell, the significance of NeurTWs is three-fold: (1) We propose a new spatiotemporal walk sampling with the exploration & exploitation trade-off to extract more diverse and expressive dynamic graph patterns than prior arts where only temporal information is considered when sampling temporal neighborhood; (2) The proposed novel continuous evolution mechanism explicitly models temporal information (i.e., time intervals between temporal events) while previous works usually failed; (3) We explore the possibility to optimize dynamic graph neural networks with contrastive learning. Thus, we believe the contributions of our work are clear and significant.
>
> - We thank the reviewer for pointing out an issue with our complexity analysis in Appx. B3. Following the assumptions made in the complexity analysis in Appx. B3, we analyze the upper bound time complexity of walk anonymization based on a simple brute-force implementation. Firstly, we identify unique nodes among $C$ temporal walks with the length $l$ by traversing all nodes, which has the complexity of $\mathcal{O}(Cl)$. Then, we can acquire the positional encoding of each node similarly with the complexity of $\mathcal{O}(Cl)$. Thus, the total time complexity of the walk anonymization is $\mathcal{O}(BCl)$ for $B$ temporal interactions in a batch. On this basis, the time complexity of our method is $\mathcal{O}(K(BC(k_s + l) + lFd_h^{2}))$. For more details, please refer to our revised Appx. B3.
>
> - Regarding the last weakness and the first question of how our continuous evolution can deal with large time intervals, our method tactfully addresses this challenge with a normalization trick. Accordingly, we have revised the Sec. 4.5 and Appx. B3 to discuss this in details.
> Indeed, some time intervals between temporal events in a real-world CTDG may be very large and make solving Eq. 8 intractable. Towards this, a good practice is to scale these time intervals without violating the inductive biases that the relative differences between time intervals reflect essential temporal information. On this basis, several transformations can be applied, such as min-max normalization, logarithmic transformations, or even divided by a large constant. In practice, we empirically find that using the base 10 logarithmic transformation is sufficient for our method to achieve satisfactory performances on four CTDG datasets. Although logarithmic transformations may skew the variances between large time intervals, we conjecture that the key is retaining the relative differences between small and large intervals to reveal critical temporal clues when modeling dynamic graphs.
>
> - For the question of handling CTDGs with a large average number of spanned temporal neighbors, we consider two solutions to alleviate the computational burden: Random and knowledge-based filtering. Although the proposed spatial-biased sampling induces higher time complexity in proportion to the number of spanned temporal neighbors $k_s$, it is practical to limit $k_s$ by selecting a set of candidates to explore. Specifically, we may choose these candidates randomly or based on prior knowledge, e.g., a more recent temporal neighbor may be more informative. A concrete example is the Enron dataset with a higher average temporal node degree compared to other datasets; we empirically find that limiting the number of first- and second-order neighbors with small upper bounds (e.g., $65 \times 5$ and 8) in a temporal-biased manner does not affect the performances of our method significantly. Corresponding revisions are made in Appx. D4 for better readability.
>
> - We have carefully corrected the pointed issue in Fig. 3. Please refer to the revised paper for details.

---

### Official Review · Reviewer_zmsk · 2022-07-11

**Rating:** 7
**Confidence:** 4
**Soundness:** 3 good
**Presentation:** 3 good
**Contribution:** 3 good

**Summary:**

This paper proposes a method based on the concepts of temporal walks and neural ordinary differential equations to learn effective node representations on continuous-time dynamic graphs. The main contributions of this work are in three aspects. Firstly, it proposes a more effective way to sample temporal walks from dynamic graphs. Secondly, it learns continuous latent dynamics of temporal walks, showing significant performance gains (Tab. 3). Thirdly, it borrows the idea of contrastive learning to optimize the dynamic graph learning problems. Compared with the existing methods, especially for its predecessor CAWs, the proposed method shows significantly better performances on downstream tasks.

**Questions:**

1.	How is the spatial information handled in the proposed neural temporal walk encoding method?

2.	Why have the authors not experimented on the widely adopted Wikipedia and Reddit datasets used in TGAT, TGN, and CAWs?

3.	What are the connections and comparisons between the random walk-based methods (e.g., NeurTWs) and message passing-based methods (e.g., TGAT and TGN)? In other words, what is the significance between NeurTWs and those local aggregation methods?


**Limitations:**

Yes. The authors have addressed the limitations of their work.

**Strengths And Weaknesses:**

Pros:

+	In general, this paper is well-presented. The learning problem on dynamic graphs is clearly formulated, and the technical writing in Sec. 4 is easy to follow. The overall architecture design makes sense to me, where the contributions and improvements over CAWs can be easily distinguished.

+	The experimental design is straightforward and comprehensive. Firstly, the effectiveness of the proposed method can be proved compared with state-of-the-art baselines. Secondly, necessary ablation and parameter studies are also clearly presented in the paper's main text. Thirdly, I can see supplementary experiments in the appendices, including the analysis of important designs and the visualization results, which help confirm the effectiveness of the proposed method.

Cons:

+	The readability can be further improved. It requires the readers to have some background knowledge on anonymous and temporal walks to fully understand the proposed method.

+	Some description is unclear. For instance, how is the spatial information handled in the proposed neural temporal walk encoding method?

---

> ### Author Response · Authors · 2022-08-02
> **Response to Reviewer zmsk**
>
> We thank the reviewer for the valuable feedback and acknowledge our technical contributions and the effectiveness of the proposed method. We address the concerns raised by the reviewer as follows.
>
> - Regarding the argument that it requires readers to have background knowledge on temporal walks and walk anonymization to understand the proposal, we respectfully disagree. The formal definitions of temporal walks and walk anonymization are given in Sec. 4.1. On this basis, we present the detailed walk sampling and anonymization methodology in the following section and Fig. 3. For motivations behind, we discuss them in the first two paragraphs in Sec. 1. A clear discussion of prior arts is also given in Sec. 2, as well as the connections between them and our method. Thus, we believe that the context of temporal walks and walk anonymization has been clearly provided for readers to understand NeurTWs.
>
> - **Q1:** How is the spatial information handled in NeurTWs?
> **Response:** The structural information is retained via the walk sampling and anonymization, where the later step replaces raw node identities with positional encodings. More details are in Sec. 4.2. In a nutshell, we consider characterizing a temporal node in a CTDG with its surrounding contexts (i.e., dynamic graph motifs). The walk sampling extracts temporal walks, and the anonymization transforms them into dynamic graph motifs where the structural information is well-preserved. After this, the proposed encoding method further considers the temporal information, allowing the spatiotemporal dynamics under each motif to be well-modeled to learn reliable time-aware node embeddings.
>
> - **Q2:** Why not experiment on Wikipedia and Reddit datasets?
> **Response:** We have not experimented on Wikipedia and Reddit because the selected evaluation metrics (i.e., AUC and AP) are relatively high among those baselines (e.g., both are close to 1.0 in CAWs). Thus, we select three more challenging real-world CTDGs (in terms of the downstream task performances among baselines) with and without attributive information to verify the effectiveness of NeurTWs. Also, we have provided additional experimental results on the MOOC dataset in our revision to better demonstrate our method's effectiveness.
>
> - **Q3:** What are the comparisons between NeurTWs and local aggregation methods?
> **Response:** On the basis of recent advances of dynamic graph neural networks, there are two types of methods to learn time-aware node embeddings on CTDGs: Node-based and subgraph-based.
>     - Compared with node-based methods (e.g., TGN and TGAT), subgraph-based approaches (e.g., NeurTWs and CAWs) show a better ability to capture essential dynamic graph motifs via temporal walks in facilitating downstream tasks. As discussed in CAWs, local aggregation (node-based) methods may fail to learn accurate structural dynamics, especially on unattributed dynamic graphs. In contrast, our approach does not necessarily rely on rich node/edge attributes to model CTDGs, showing promising results in both attributed and unattributed scenarios. We discussed this in Sec. 5.2.
>     - Most node-based (as well as subgraph-based) methods learn time-aware node embeddings by aggregating the messages from informative temporal neighbors. However, previous research only considers the temporal information when measuring the importance of temporal neighborhood, where the structural and traversal properties are completely ignored. We discussed this issue in the second paragraph of the introduction and are motivated to address it with the proposed spatiotemporal walk sampling.
>     - Node-based methods aggregate the information from irregularly-sampled temporal neighbors with a time encoding trick proposed in TGAT, in which we argue that the temporal information has not been modeled explicitly (as discussed in the second paragraph of the introduction). Our proposal, in contrast, designs a novel walk encoding method to learn from spatial and temporal information explicitly.
>     - Different from most node-based (as well as subgraph-based) methods, we introduce a harder contrastive pretext task in optimizing dynamic graph neural networks, where the supervision signals from temporal interactions are enriched effectively (see the penultimate paragraph in Sec. 1).

---

> > ### Comment · Reviewer_zmsk · 2022-08-07
> > **Feedback**
> >
> > Thanks to the authors for their detailed reply. My concerns are clarified.

---

### Official Review · Reviewer_qteb · 2022-07-11

**Rating:** 5
**Confidence:** 4
**Soundness:** 2 fair
**Presentation:** 3 good
**Contribution:** 2 fair

**Summary:**

The paper proposes a new approach for representation learning on temporal graphs (or CTDGs). The proposal can be viewed as an extension of the walk sampling scheme from CAW [1], modified to also count on node degree and time traversal information. The paper also employs a NeuralODE-based approach aiming to better leverage fine-grained temporal information. Results on three benchmark datasets show the effectiveness of the proposal.


**Questions:**

- In the Introduction, the paper says “[..] studies in dynamic graph modelling extract expressive patterns manually […]” and cites [1]. Although [1] uses triadic closure as a motivation, the design in [1] can capture a large collection of structures/motifs. Why is this regarded as manual motif extraction?
- “[..] time encoding hurts model performance because temporal dependencies are modelled implicitly”. What is the basis for this strong claim?
- In Definition 1: $i \in \mathbb{N}^+$ means that $i$ lies in the natural numbers without zero, which conflicts with $0 \leq i$. Also, by definition of $G_{\cdot, t}$, no pair $(\cdot, t_i)$ can belong to ${G}_{\cdot, t_i}$. This definition deserves some polishing.
- What can continuous evolution bring in terms of expressive power that sequence-based models (e.g., RNNs) cannot?
- The chosen datasets are small. It would be helpful to report the performance of NeurTW on the Reddit and MOOC datasets.
- Is the proposed model able to deal with (node-level and edge-level) deletion events? How would so?
- Can the model be applied to node-level prediction tasks? How could we obtain node embeddings using NeurTW?
- In the ablation study Ab.4, what is done to replace the continuous evolution procedure? Is the temporal information completely disregarded?
- It would be helpful to report a time comparison between NeurTW vs. TGN and CAW.

**Typos:**
- Line 165: "a nodes u"
- Line 162: "to allows"
- Line 126: "Specifically, Given"
- Line 187: "with the length $l$"
- Line 234: "We details"

[1] CAW: https://arxiv.org/pdf/2101.05974.pdf

**Limitations:**

The authors do not discuss the limitations of their work.

**Strengths And Weaknesses:**

Overall, the paper is easy to follow despite some typos and minor technical errors. I like the simplicity of the proposed method and the fact it leads to promising results. Also, the paper discusses a relevant topic in AI/ML. However, I would rank novelty as limited: modification of existing approaches (sampling strategy of CAW) combined with previous strategies (ODE-based solvers, contrastive loss) and evaluated using a standard setup. Also, no theoretical insights or claims are established in the paper. The proposal seems to incur a computation burden on a model that is already slow (CAW, [1]). Last, the paper only considers three benchmarks as opposed to six in [1].

Strengths:
- Promising results
- Simplicity

Weaknesses:
- Lack of a more principled motivation behind the proposed design choices
- Limited evaluation setup
- Limited technical novelty

---

> ### Author Response · Authors · 2022-08-02
> **Response to Reviewer qteb (Part 1)**
>
> We thank the reviewer for the valuable comments on our paper and appreciating the simplicity and effectiveness of our proposed method. We especially thank the reviewer for pointing out some issues in writing (e.g., Definition 4.1.1) and experimental and implementation details (e.g., datasets, downstream tasks, and how to deal with deletion events), which help improve the presentation of our paper.
>
>
> Below we first address some main comments.
>
> - **Motivation and novelty:** Although path-based approaches have recently demonstrated strong capability in CTDG representation learning, our research is motivated to provide thoughtful insights into overcoming their fundamental limitations from various aspects: (1) In light of the advances in static graph modeling, we first propose a new sampling algorithm that emphasizes complementing the structural and traversal information overlooked in existing works when retrieving patterns from temporal networks; (2) The proposed novel temporal walk encoding first makes it possible to directly and effectively (Tab. 6 in Appx. D2) learn from (large-gapped) time intervals between irregularly-sampled temporal nodes without relying on any crafted surrogates (e.g., encoding timestamps with random Fourier features); (3) Ubiquitous interactions in temporal networks provide essential information to supervise the training of dynamic graph neural networks, while most existing works fail to fully utilize such information. Towards this, we first consider enriching supervision signals with a new noise-contrastive objective. See also our discussion in response to Q3 asked by reviewer zmsk, where the significance between our method and a broader range of dynamic graph neural networks is discussed.
>
> - **Evaluation:** We follow a standard experimental setup adopted in many prior works, and a fair comparison with baseline models is only enabled by adopting a standard evaluation protocol. We have conducted additional experiments suggested by the reviewer and provided the results and discussions in the revised paper. As can be seen in the updated Tab. 2 (main body) and Tab. 5 (Appx. D1), our model outperforms baseline models significantly on most of cases on the MOOC dataset. See our response to Q5 for an explanation of why not experiment on Wikipedia and Reddit datasets.
>
> - **Additional computational burden:** We have presented a theoretical time complexity analysis in Appx. B3 and a detailed empirical analyses in Appx. D4. As we have acknowledged in the paper, while our proposed spatiotemporal-biased sampling induces higher time complexity, we show that it helps extract more diverse and expressive dynamic graph motifs than existing temporal-biased sampling methods. Moreover, as we discussed in D4, a number of strategies can be employed to reduce time cost. These include limiting the number of spanned temporal neighbors, increasing the step size and limiting the number of negatives. We note that these strategies explicitly trade model precision for speed. See our revised Fig. 3 and related discussions in Appx. D4 for more details.
>
>
> We then address the detailed concerns raised by the reviewer as follows.
>
> ### Q1. Why is CAWs regarded as a method with manual motif extraction?
> Sorry for the confusion. In the first paragraph of the introduction, we emphasize the fact that traditional studies in dynamic graph modeling usually extract dynamic graph patterns manually based on the discussion in CAWs, which answers why we cite it after this sentence. In other words, we are not saying that CAWs extract temporal graph motifs manually.
>
> ### Q2. '.. time encoding hurts model performance because temporal dependencies are modelled implicitly'. What is the basis for this strong claim?
> We argue that most of the existing dynamic graph modeling approaches fail to model temporal information (i.e., time intervals between temporal events) explicitly. Among them, a standard trick is concatenating latent states with the encoding of time intervals to enable MP-GNNs or sequence models aggregating information from irregularly-sampled temporal neighbors (e.g., TGAT and CAWs), where the temporal information has not been explicitly leveraged in their modeling process. Continuous-time models with latent states defined at all times, on the other hand, better match reality and have been shown to be more effective in some research fields [1],[2],[3]. In light of this, we propose the novel walk encoding on dynamic graphs to properly and explicitly define the latent states between observations (i.e., temporal nodes), where the detailed study in Appx. D2 empirically demonstrates that our approach has a better modeling precision than the time encoding-based variant.
>
> [1] GRU-ODE-Bayes: Continuous modeling of sporadically-observed time series (NIPS 2019)
>
> [2] Latent ODEs for Irregularly-Sampled Time Series (NIPS 2019)
>
> [3] Neural Controlled Differential Equations for Irregular Time Series (NIPS 2020)

---

> > ### Author Response · Authors · 2022-08-02
> > **Response to Reviewer qteb (Part 2)**
> >
> > ### Q3. In Definition 1: $i \in \mathbb{N}^{+}$ means that $i$ lies in the natural numbers without zero, which conflicts with $0 \leq i$. Also, by definition of $G_{\cdot, t}$, no pair $(\cdot, t_i)$ can belong to $G_{\cdot, t_i}$. This definition deserves some polishing.
> >
> > Thank you for pointing out these issues. We have corrected $i \in \mathbb{N}^+$ with $i \in \mathbb{N}$, and revised the definition of temporal walk as $W = \\{ (w_i, t_i) \\ | \\ i \in \mathbb{N},\\  0 \leq i \leq l, \\  t_0 > t_1 > \cdots > t_l, \\  (\\{w_{i}, w_{i-1}\\}, t_i) \in \mathcal G_{w_{i-1}, t_{i-1}}\\  \text{for}\\  i \geq 1\\}$.
> >
> > ### Q4. What can continuous evolution bring in terms of expressive power that sequence-based models (e.g., RNNs) cannot?
> >
> > Most of sequence models (e.g., RNNs) cannot directly process sequential data with irregular gaps. For example, RNNs can be interpreted as discrete approximations to some functions of dynamic systems, while such a discretization typically breaks down if the data is irregularly observed [1]. In our experiments, we find that using a standard sequence model hurts the performance dramatically (see Tab. 6 in Appx. D2) because the valuable temporal information has been discarded by assuming the time intervals between temporal nodes are the same. In contrast, our proposal preserves such information by explicitly integrating over multiple interaction time intervals to learn the latent spatiotemporal dynamics, which is essential to obtain expressive temporal node embeddings (Ab. 0 vs. Ab. 4 in Tab. 3).
> >
> > [1] Neural Controlled Differential Equations for Irregular Time Series (NIPS 2020)
> >
> >
> > ### Q5. The chosen datasets are small. It would be helpful to report the performance of NeurTWs on the Reddit and MOOC datasets.
> >
> > We have not experimented on Wikipedia and Reddit because the selected evaluation metrics (i.e., AUC and AP) are relatively high among those baselines (e.g., both are close to 1.0 in CAWs). See also our response to Q2 asked by reviewer zmsk. In response to the concern of dataset size, we provide the supplementary experimental results on MOOC as follows:
> >
> > |                  |          Transductive          |                                |       Inductive (New-Old)      |                                |       Inductive (New-New)      |                                |
> > |:----------------:|:------------------------------:|--------------------------------|:------------------------------:|--------------------------------|:------------------------------:|--------------------------------|
> > |                  |               AUC              |               AP               |               AUC              |               AP               |               AUC              |               AP               |
> > |       JODIE      |       0.6815 $\pm$ 0.014       |       0.6386 $\pm$ 0.021       |       0.6304 $\pm$ 0.006       |       0.6914 $\pm$ 0.010       |       0.8243 $\pm$ 0.007       |     **0.8263 $\pm$ 0.015**     |
> > |       DyRep      |       0.6195 $\pm$ 0.018       |       0.5723 $\pm$ 0.005       |       0.5504 $\pm$ 0.010       |       0.5633 $\pm$ 0.014       |       0.5288 $\pm$ 0.021       |       0.4484 $\pm$ 0.047       |
> > |       TGAT       |       0.6750 $\pm$ 0.035       |       0.6704 $\pm$ 0.032       |       0.6410 $\pm$ 0.024       |       0.6408 $\pm$ 0.014       |       0.6365 $\pm$ 0.014       |       0.6324 $\pm$ 0.012       |
> > |        TGN       | $\underline{0.7703 \pm 0.032}$ | $\underline{0.7347 \pm 0.036}$ |       0.6968 $\pm$ 0.008       |       0.7271 $\pm$ 0.002       |       0.6448 $\pm$ 0.053       |       0.6547 $\pm$ 0.065       |
> > |       CAWs       |       0.6984 $\pm$ 0.053       |       0.7011 $\pm$ 0.062       |       0.7479 $\pm$ 0.023       |       0.7602 $\pm$ 0.022       |       0.7558 $\pm$ 0.036       |       0.7688 $\pm$ 0.037       |
> > |      $\texttt{NeurTWs}$     |     **0.7756 $\pm$ 0.031**     |     **0.7593 $\pm$ 0.027**     |     **0.7822 $\pm$ 0.004**     |     **0.7730 $\pm$ 0.006**     |     **0.8329 $\pm$ 0.010**     | $\underline{0.8251 \pm 0.017}$ |
> > | $\texttt{NeurTWs} \dagger$ |       0.7470 $\pm$ 0.028       |       0.7228 $\pm$ 0.033       | $\underline{0.7772 \pm 0.006}$ | $\underline{0.7617 \pm 0.008}$ | $\underline{0.8302 \pm 0.007}$ |       0.8067 $\pm$ 0.013       |

---

> > > ### Author Response · Authors · 2022-08-02
> > > **Response to Reviewer qteb (Part 3)**
> > >
> > > ### Q6. Is the proposed model able to deal with (node-level and edge-level) deletion events? How would so?
> > >
> > > Although we define a CTDG as a set of temporal interactions, our method is not likely to be affected by deletion events described in TGN. We first discuss why dynamic graph neural networks with the node memory mechanism (e.g., TGN) must handle deletion events properly. In these methods, the time-aware embedding of a node is learned based on its current memory (and features) and the aggregated temporal neighbors' current memories (and features). Since a node memory is the summary of the historical events this node participated in, deletion events related to those historical events may not result in the change of its memory, which thus affects the modeling process. In contrast, our method does not rely on node memories when learning time-aware node embeddings. In NeurTWs, given a node at time $t$, we may sample a set of temporal subgraphs around it to represent this node. The deletion events at time $t$ related to the temporal neighbors of this node will not affect the embedding calculation of it at time $t+1$ because: (1) We will resample surrounded patterns to characterize this node; (2) The historical embeddings of this node and its temporal neighbors (at time $t+1$) will not affect its embedding calculation at time $t+1$.
> > >
> > > ### Q7. Can the model be applied to node-level prediction tasks? How could we obtain node embeddings using NeurTWs?
> > >
> > > The proposed method supports node-level tasks in principle. In NeurTWs, we learn time-aware node embeddings where the model is supervised by the temporal interactions. To conduct node-level tasks (e.g., node classification), we may first choose the unitary anonymization and pre-train the model with temporal interactions. After this, we can freeze the pre-trained model to generate temporal node embeddings and train a downstream decoder to perform a specific node-level task on CTDGs, as in TGAT and TGN.
> > >
> > > ### Q8. In the ablation study Ab.4, what is done to replace the continuous evolution procedure? Is the temporal information completely disregarded?
> > >
> > > ‘’w/o continuous evolution'' denotes disabling Eq. 8, which degrades the encoding method to GRU without considering the temporal information. See also our response to Q4.
> > >
> > > ### Q9. It would be helpful to report a time comparison between NeurTWs vs. TGN and CAWs.
> > >
> > > Thank you for the valuable suggestion. We have provided supplementary experimental results and related discussion in Appx. D4. As discussed above, although the proposed method induces higher time complexity than those baselines, each of our core components (e.g., walk sampling and encoding) allows trading model precision for speed. See our revised Appx. D4 for detailed discussions.

---

> > > > ### Comment · Reviewer_qteb · 2022-08-09
> > > > **Response to authors**
> > > >
> > > > Thanks for your reply. I appreciate the effort to report additional results on MOOC.
> > > >
> > > > My concerns have been only partially addressed. In particular,
> > > >
> > > > 1. the high accuracy of CAW on Reddit and Wikipedia is not a strong reason to disregard these datasets. Also, the CAW paper considers other datasets that are not considered here.
> > > >
> > > > 2. if I understood correctly, you somehow argue that simply treating deletion events as if they have never happened is a better way to deal with them than TGN's approach. I am not convinced. In addition, this would require recomputing sampling probabilities.
> > > >
> > > > 3. there is no evidence that the unitary anonymization approach (obtaining walks from a single node/endpoint!?) should work for node-level events. It would be interesting to see how this would compare against TGNs, for instance.
> > > >
> > > > 4. The link for the code seems broken. Could you check?
> > > >
> > > > That being said, given the promising results now backed by an additional experiment, I decided to raise my score from 4 to 5.

---

> > > > > ### Author Response · Authors · 2022-08-09
> > > > > **Thanks for your acknowledgement**
> > > > >
> > > > > We thank the reviewer for the follow-up comments and are delighted that some of your concerns have been addressed. Here, we provide brief remarks in response to the listed questions/concerns, and we hope they are somewhat helpful.
> > > > >
> > > > > > The high accuracy of CAW on Reddit and Wikipedia is not a strong reason to disregard these datasets. Also, the CAW paper considers other datasets that are not considered here.
> > > > >
> > > > > Even though we suspect that the performances (in terms of AUC and AP) between different methods on Wikipedia and Reddit may not be pronounced under a standard setting (e.g., data split and node masking ratios), we are happy to add experiments on these two datasets in the follow-up revisions to address your concern if it is necessary.
> > > > >
> > > > > > If I understood correctly, you somehow argue that simply treating deletion events as if they have never happened is a better way to deal with them than TGN's approach. I am not convinced. In addition, this would require recomputing sampling probabilities.
> > > > >
> > > > > We do not deny the existence of deletion events in CTDGs. We argue that the approaches with node memories (e.g., TGN) need to explicitly and carefully handle deletion events because these events may lead to the change of node memories as they are summaries of historical interactions, which directly relates to the calculation of node embeddings at specific timestamps. Our method, in contrast, is not in an incremental manner (we mean, not with the help of node memories) when calculating time-aware node embeddings. In other words, our method considers the “instantaneous state” of a CTDG with interactions before $t$ when calculating the node embedding at $t$. Thus, our approach models so-called deletion events naturally with the cost of recomputing sampling probabilities.
> > > > >
> > > > > > There is no evidence that the unitary anonymization approach (obtaining walks from a single node/endpoint!?) should work for node-level events. It would be interesting to see how this would compare against TGNs, for instance.
> > > > >
> > > > > Indeed, a node-level task (e.g., dynamic node classification) may also be worth considering. We are happy to provide a related study in the following revisions. We conjecture that unitary anonymization works for node-level tasks based on the following facts:
> > > > >
> > > > > - Binary anonymization aims to establish the connections between walks around a pair of target nodes when generating dynamic graph motifs, while unitary anonymization focuses on a single target node.
> > > > >
> > > > > - We focus on the entanglement between a pair of nodes in edge-level tasks, while we are only interested in characterizing a temporal node with its surrounded patterns when performing node-level tasks.
> > > > >
> > > > > - Our experiments show that unitary anonymization is already sufficient for our method to achieve satisfactory performances in most cases, which makes our solution (mentioned in our previous response to Q7) feasible.
> > > > >
> > > > > Of course, we are happier to justify this with experimental evidence in the following revisions, as the reviewer suggested.
> > > > >
> > > > > > The link for the code seems broken. Could you check?
> > > > >
> > > > > We thank the reviewer for reminding us about this issue. We are using Anonymous Github to host the repository, and we have found their internal server errors caused this technical issue. We will work on it to ensure the code link back online as soon as possible.
> > > > >
> > > > > Although the time is limited, we enjoyed the discussion with the reviewer, and we would like to thank the reviewer again for the valuable time and insightful comments.

---

> ### Author Response · Authors · 2022-08-09
> **General Comment to Reviewer qteb**
>
> We thank the reviewer for the insightful comments and valuable suggestions.
>
> It would be great if we could hear from you to discuss whether we addressed your questions and concerns since the author-reviewer discussion is ending soon.

---

### Author Response · Authors · 2022-08-02
**General Response to All Reviewers**

We sincerely thank all the reviewers for their time and insightful comments. We are excited that all of the reviewers acknowledged the effectiveness and simplicity of the proposed method, and we appreciate that they find our paper is well motivated (Reviewers *zmsk* and *avBg*), original (Reviewers *zmsk* and *avBg*), technically solid (Reviewer *zmsk*), and well presented (Reviewers *qteb*, *zmsk*, and *avBg*).

To the best of our efforts, we have provided detailed comments to address the concerns raised by each reviewer. Meanwhile, we have carefully revised the paper and appendices where the main modifications are highlighted in red. Specifically, the main revisions we made are as follows.

- We have added experimental results (Tab. 2 in the main text and Tab. 5 in Appx. D1) and related discussions on the MOOC dataset.
- We have revised Definition 4.1.1 and the complexity analysis in Appx. B3.
- We have added a discussion in Appx. B3 related to time interval normalization.
- We have added the experimental results of time comparison (Fig. 3(d) in Appx. D4) and related discussions in Appx. D4.
- We have corrected all typos and presenting issues pointed out by the reviewers.

---

### Meta-Review · Area_Chair_gNqU · 2022-08-30

**Recommendation:** Accept
**Confidence:** Certain

**Metareview:**

The paper proposes a method based on the concepts of temporal walks and neural ordinary differential equations to learn effective node representations on continuous-time dynamic graphs. All the reviewers are positive about the paper and the rebuttal/discussion helped clearing out the concerns they had.

**Award:**

No

---

### Decision · Program_Chairs · 2022-09-14

Accept